# Effects of climate and land-use changes on fish catches across lakes at a global scale

Yu-Chun Kao ⓘ et al.[#]

Globally, our knowledge on lake fisheries is still limited despite their importance to food security and livelihoods. Here we show that fish catches can respond either positively or negatively to climate and land-use changes, by analyzing time-series data (1970–2014) for 31 lakes across five continents. We find that effects of a climate or land-use driver (e.g., air temperature) on lake environment could be relatively consistent in directions, but consequential changes in a lake-environmental factor (e.g., water temperature) could result in either increases or decreases in fish catch in a given lake. A subsequent correlation analysis indicates that reductions in fish catch was less likely to occur in response to potential climate and land-use changes if a lake is located in a region with greater access to clean water. This finding suggests that adequate investments for water-quality protection and water-use efficiency can provide additional benefits to lake fisheries and food security.

[#]A full list of authors and their affiliations appears at the end of the paper.

Lakes, natural and man-made, are important reserves of accessible freshwater and are frequently managed for provisioning ecosystem services[1], such as drinking water, irrigation, hydropower generation, and fisheries[2]. Consequently, biota in lake ecosystems could be subject to stressors associated with human activities, such as water withdrawal, floodplain development, exotic species introduction, and overexploitation[3]. Two important sources of anthropogenic stressors are climate and land-use changes, which may alter a lake ecosystem directly through changing water temperature (WT) and water level (WL), and indirectly through the balance between inputs and outputs of nutrients and sediments, as well as changes in food-web dynamics[4,5].

Among lake ecosystem services, fisheries are especially vulnerable to environmental changes because fish are ectothermic and fish distribution is usually limited by structural[6], thermal[7], and chemical[8] barriers. Environmental changes driven by climate and land-use changes have been linked to major shifts in fish catches (CATCHs) and species composition in many lakes around the world, such as lakes Erie[9], Kinneret[10], Naivasha[11,12], Peipsi[13], and Victoria[14], as well as the collapse of Aral Sea fisheries[15]. Reduced lake CATCHs caused by climate and land-use changes can threaten food security and livelihoods of millions of people worldwide, especially in impoverished countries where rural poor communities may not have appropriate alternative sources of animal protein and employment opportunity[16].

Due to their protracted and compounding nature, understanding the effects of climate and land-use changes on lake CATCHs requires long-term data, which may include climate and land-use data at a spatial scale of the whole catchment (or drainage basin) and environmental, biological, and fishery data at a spatial scale of the whole lake. This intensive data requirement has restricted assessments of how lake environment and lake CATCHs respond to climate and land-use changes to lakes in a few data-rich regions, mostly located in North America and Europe[17,18]. As a result, given the large number (>1.43 million) of lakes[19] and their diversity in size, productivity, species composition, and fisheries[6], our current understanding of how changing climate and land-use influences lake CATCHs is very limited at a global scale and is generally lacking for lakes in Africa and Asia where CATCHs are the largest (Supplementary Fig. 1).

The goal of this study is to understand how climate and land-use changes affect lake CATCHs at a global scale. We use a Bayesian networks modeling approach[20] to analyze time-series data for the 31 study lakes (Fig. 1) over the period 1970–2014. We choose this modeling approach because it can minimize negative effects of data limitations on the statistical power of our analysis, by accounting for our prior understanding of multi-level, causal processes that could underlie changes in CATCHs in response to changes in climate and land use across lakes. Our Bayesian networks model (BNM) represents the hypothesized causal processes by expressing them as conditional probabilistic relationships between model variables at multiple levels: between climate and land use (operating at a catchment scale) and lake environment[21–26], and between lake environment and (total) fish biomass[27–33] and thus (total) CATCH[34], as shown in Fig. 2. Due to the regional difference in data availability, our study lakes and study period are selected to ensure that we can include approximately equal numbers of study lakes in Africa, the Americas, Asia, and Europe and in both tropic and temperate regions (Fig. 1). The study lakes are diverse in terms of both socio-economic environment and hydrogeomorphology (Supplementary Data 2). However, most study lakes in Europe and the Americas have a better socio-economic environment than the study lakes in Asia and Africa. The diversity of these lakes allows us to identify the characteristics of lakes where CATCHs can be vulnerable to climate and land-use changes, which has implications for identifying vulnerable lake fisheries around the globe.

We conduct three analyses in this study. First, we determine how the climate and land-use drivers affect CATCHs across the 31 study lakes in 1970–2014 by estimating coefficients of our BNM. Second, by implementing the resulting BNM, we run simulations to assess the change in each of climate and land-use drivers associated with a (simulated) 25% decrease in CATCH from the 1970–2014 median (hereafter, a 25% catch decrease) in each lake. Third, we conduct a correlation analysis to identify whether any socio-economic characteristic(s) associated with the catchment or hydrogeomorphological characteristic(s) of the lake corresponded to the magnitudes of changes in any of the climate and land-use drivers that were associated with a 25% catch decrease. To improve clarity, Fig. 3 presents a flowchart that summarizes the procedures of these three analyses and Table 1 provides a list of symbols used in the development of BNM.

We demonstrate that climate and land-use changes can have strong effects on CATCHs across lakes at a global scale, but the effects can be either positive or negative. The subsequent

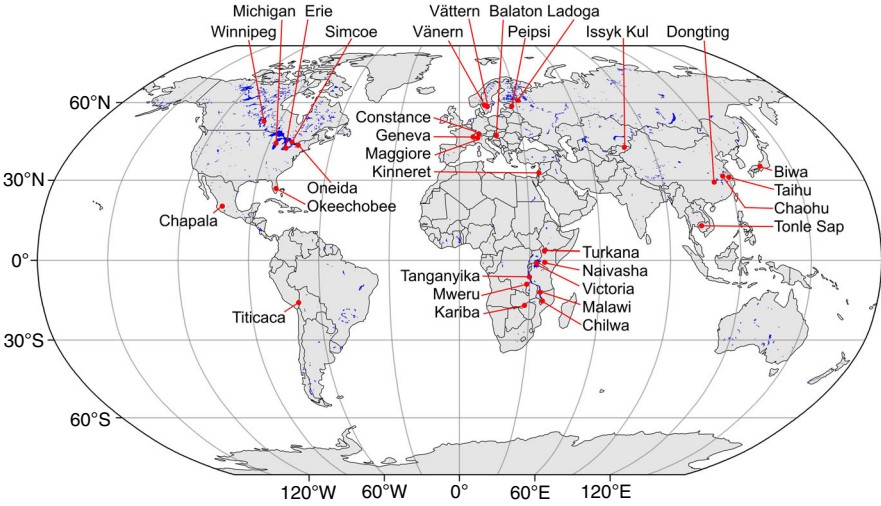

**Fig. 1 Global distribution of the 31 study lakes.** The map was generated by using R[58] and the R package rworldmap[62] to plot the centers of study lakes (as given in Supplementary Data 2) on a base world map, of which layers, including country borders, continent outlines, global lakes, and latitude–longitude grids, were obtained from a public domain map dataset Natural Earth (www.naturalearthdata.com).

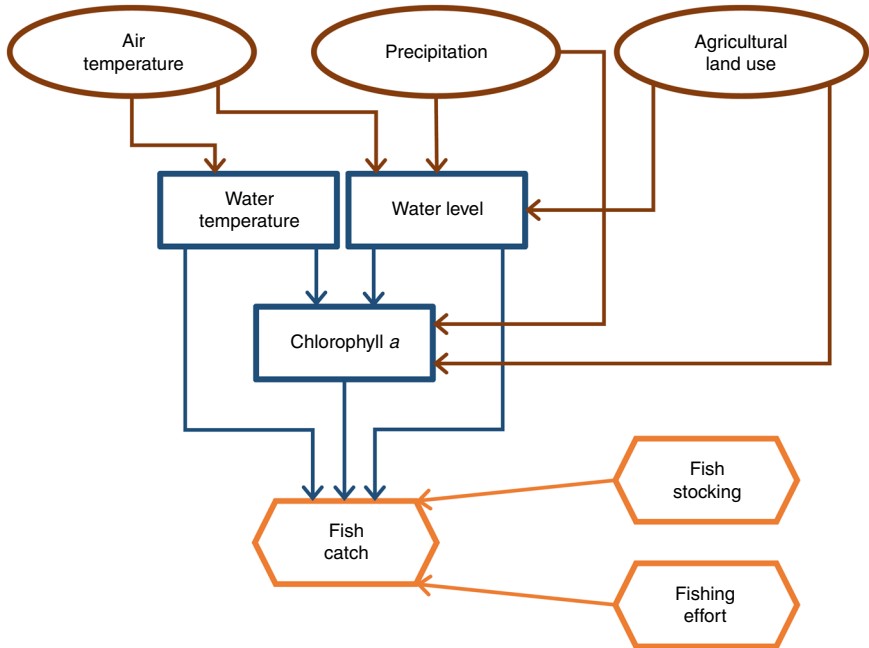

**Fig. 2 Hypothesized causal processes represented in the Bayesian networks model.** Variables associated with catchment climate and land use are in oval-brown boxes. Variables associated with lake environment are in blue-rectangular boxes. Variables associated with fish catch, fish stocking, and fishing effort are in orange-hexagon boxes.

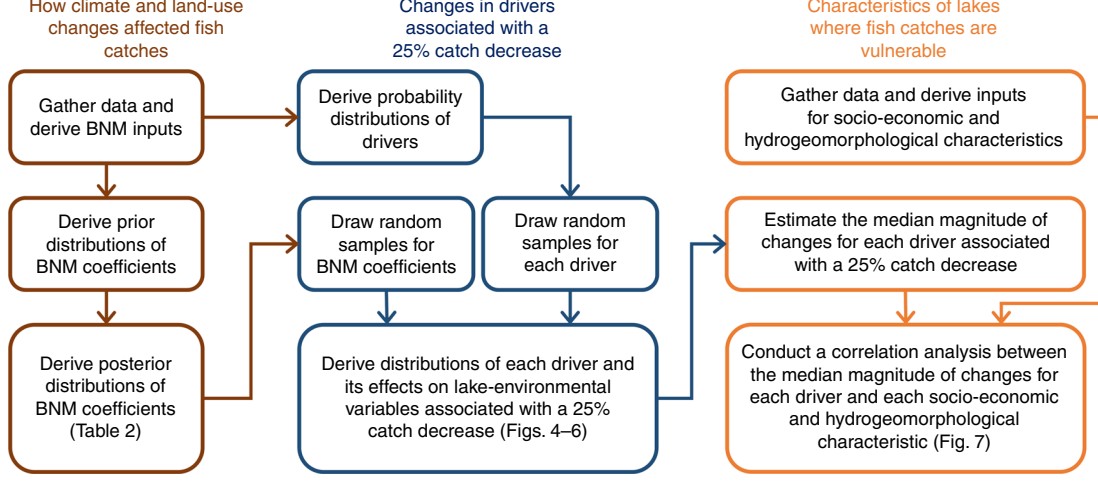

**Fig. 3 Summary flowchart for the procedures of three analyses conducted in this study.** BNM Bayesian networks model.

correlation analysis shows that a lake located in a region with greater access to clean water less likely experiences a substantial decrease in CATCH in response to potential climate and land-use changes. These results suggest that investments made for water-quality protection and water-use efficiency can also provide benefits to lake fisheries and food security.

## Results

**How climate and land-use changes affected CATCHs.** The BNM showed that climate and land-use changes could result in either increases or decreases in CATCHs across the 31 study lakes (Table 2). These results accounted for fish stocking and fishing effort (EFF), which both could have strong effects on CATCHs[34–36] (Table 2). Based on the 75% one-tailed credible intervals (CIs) of BNM coefficients, the effects of a climate or land-use driver (one of air temperature (AT), precipitation (PRE),

and agricultural land use (LUag) in this study) on lake-environmental factors (WT, WL, and primary productivity) were more consistent in directions (positive or negative) across lakes than the consequential changes in a lake-environmental factor on CATCHs across lakes. Specifically, four of the six BNM variable pairs between the climate land-use drivers and lake-environmental factors (brown arrows in Fig. 2; Table 2) were consistent in the directional relationships across at least half of the 31 study lakes: AT was positively related to WT in all 31 study lakes, but negatively related to WL in 17 lakes, PRE was positively related to WL in 29 lakes, and LUag was positively related to chlorophyll a (CHL), the measure of primary productivity in this study, in 16 lakes. In contrast, none of the 5 BNM variable pairs relating lake-environmental factors to each other or to CATCH (blue arrows in Fig. 2; Table 2) had a consistent directional relationship across the study lakes, although the closest was WT being negatively related to CHL in 15 lakes.

**Table 1 List of symbols used in the development of BNM.**

| Category | Symbol | Description (unit) |
|---|---|---|
| Parameter and coefficient | $\alpha$ | Empirical coefficient that was not estimated |
| | $\beta$ | BNM coefficient, which might be expressed a function of $\alpha$ coefficients |
| | $\mu$ | Mean or predicted value |
| | $\sigma$ | Standard deviation |
| Variables associated with climate and land use | AT | Air temperature (°C) |
| | LUag | Agricultural land use (% of catchment area) |
| | LUun | Undeveloped land use (% of catchment area) |
| | PE | Potential evaporation (m) |
| | PRE | Precipitation (m) |
| | RUNOFF | Catchment runoff (m³) |
| Variables associated with lake environment | ΔWL | Change in water level (m) |
| | CHL | Concentration of chlorophyll $a$ (µg/l) |
| | WT | Water temperature (°C) |
| Variables associated with fish catch, fish stocking, and fishing effort | B | Fish biomass (kg/ha) |
| | CATCH | Fish catch (kg/ha) |
| | EFF | Fishing effort (dimensionless) |
| | Q | Fishing catchability (dimensionless) |
| | ST | Number of fish stocked (number per hectare) |

*BNM* Bayesian networks model.

**Table 2 Predicted effects between variable pairs in the BNM across lakes.**

| BNM variable pair | | Number of lakes | | |
|---|---|---|---|---|
| Predictor | Response | Positive effects | Negative effcts | Mixed effects |
| Air temperature | Water temperature | 31 | 0 | 0 |
| | Water level | 6 | 17 | 8 |
| Precipitation | Water level | 29 | 0 | 2 |
| | Chlorophyll $a$ | 9 | 7 | 15 |
| Agricultural land use | Water level | 8 | 12 | 11 |
| | Chlorophyll $a$ | 16 | 8 | 7 |
| Water temperature | Chlorophyll $a$ | 6 | 15 | 10 |
| | Fish catch | 10 | 10 | 11 |
| Water level | Chlorophyll $a$ | 13 | 8 | 10 |
| | Fish catch | 9 | 5 | 17 |
| Chlorophyll $a$ | Fish catch | 12 | 11 | 8 |
| Fish stocking | Fish catch | 4 | 1 | 3 |
| Fishing effort | Fish catch | 30 | 0 | 1 |

Effects between a variable pair were positive, negative, and mixed based on one-tailed 75% credible intervals of BNM coefficients, for which complete summary statistics are given in Supplementary Data 4. Effects of stocking on fish catch were only included in eight lakes where stocked species contributed >20% of fish catches in >10 years in the study period 1970–2014.
*BNM* Bayesian networks model.

**Changes in drivers associated with a 25% catch decrease**. Our simulations showed that a 25% catch decrease (from the 1970–2014 medians) was associated with both colder and warmer AT across the 31 study lakes (Fig. 4a), and the effects of AT on CATCHs were mostly mediated though changes in WT (Fig. 4e, h) rather than WL (Fig. 4f) or CHL (Fig. 4g). For 8 of the 31 lakes, a simulated 25% catch decrease was associated with colder AT (i.e., a positive relationship), with medians of 0.9–3.6 °C colder than the observed 1970–2014 medians (Fig. 4a). For another eight lakes, however, a simulated 25% catch decrease was associated with warmer AT (i.e., a negative relationship), with medians of 0.8–1.7 °C warmer than the observed 1970–2014 medians. For these 16 lakes, the simulated relationship between AT and WT was consistently

positive (Fig. 4b), whereas the simulated relationships between AT and WL (Fig. 4c) and between AT and CHL (Fig. 4d) could be either positive, negative, or mixed. Translating these simulated changes to effects on CATCHs revealed that the effects of WT, resulting from changes in AT, were important in 8 of these 16 lakes (i.e., leading to a >9.1% catch decrease in simulations; Fig. 4e), even though both colder and warmer WT could lead to a catch decrease. The effects of WL, resulting from changes in AT, on catch decrease were important in 2 of these 16 lakes (Fig. 4f), where warmer AT were associated with lower WL which, in turn, resulted in lower catches. The effects of CHL, resulting from changes in AT, on catch decrease were not important in any of these 16 lakes. For the remaining 15 lakes, a mixed relationship was revealed as a simulated 25% catch decrease was associated with both colder and warmer AT.

A simulated 25% catch decrease was associated with both higher and lower PRE across the study lakes (Fig. 5a), but these effects were mediated through changes in both WL (Fig. 5d, f) and CHL (Fig. 5e, f). For 10 lakes, a simulated 25% catch decrease was associated with lower PRE (i.e., a positive relationship), with medians of 15–71% lower than the observed 1970–2014 medians (Fig. 5a). For another six lakes, a simulated 25% catch decrease was associated with higher PRE (i.e., a negative relationship), with medians of 20–36% higher than the observed 1970–2014 medians. For these 16 lakes, the simulated relationship between PRE and WL was consistently positive (Fig. 5b), while the simulated relationship between PRE and CHL could be either positive, negative, or mixed (Fig. 5c). Translating these simulated changes to effects on CATCHs revealed that the effects of CHL were important in 7 of these 16 lakes (i.e., leading to a >13.4% catch decrease in simulations), but inconsistent relationships underlie this pattern: lower PRE was associated with both lower and higher CHL. The effects of WL, resulting from changes in PRE, on catch decrease were important only in 2 of these 16 lakes (Fig. 5d), where higher PRE increased WL. For the remaining 15 lakes, a mixed relationship was revealed as a simulated 25% catch decrease was associated with both lower and higher PRE.

Similarly, a simulated 25% catch decrease was associated with both increasing and decreasing LUag across the 31 study lakes (Fig. 6a), but effects of LUag on CATCHs were mostly mediated through changes in CHL (Fig. 6e, f) rather than through WL (Fig. 6d, f). For nine lakes, a simulated 25% catch decrease was

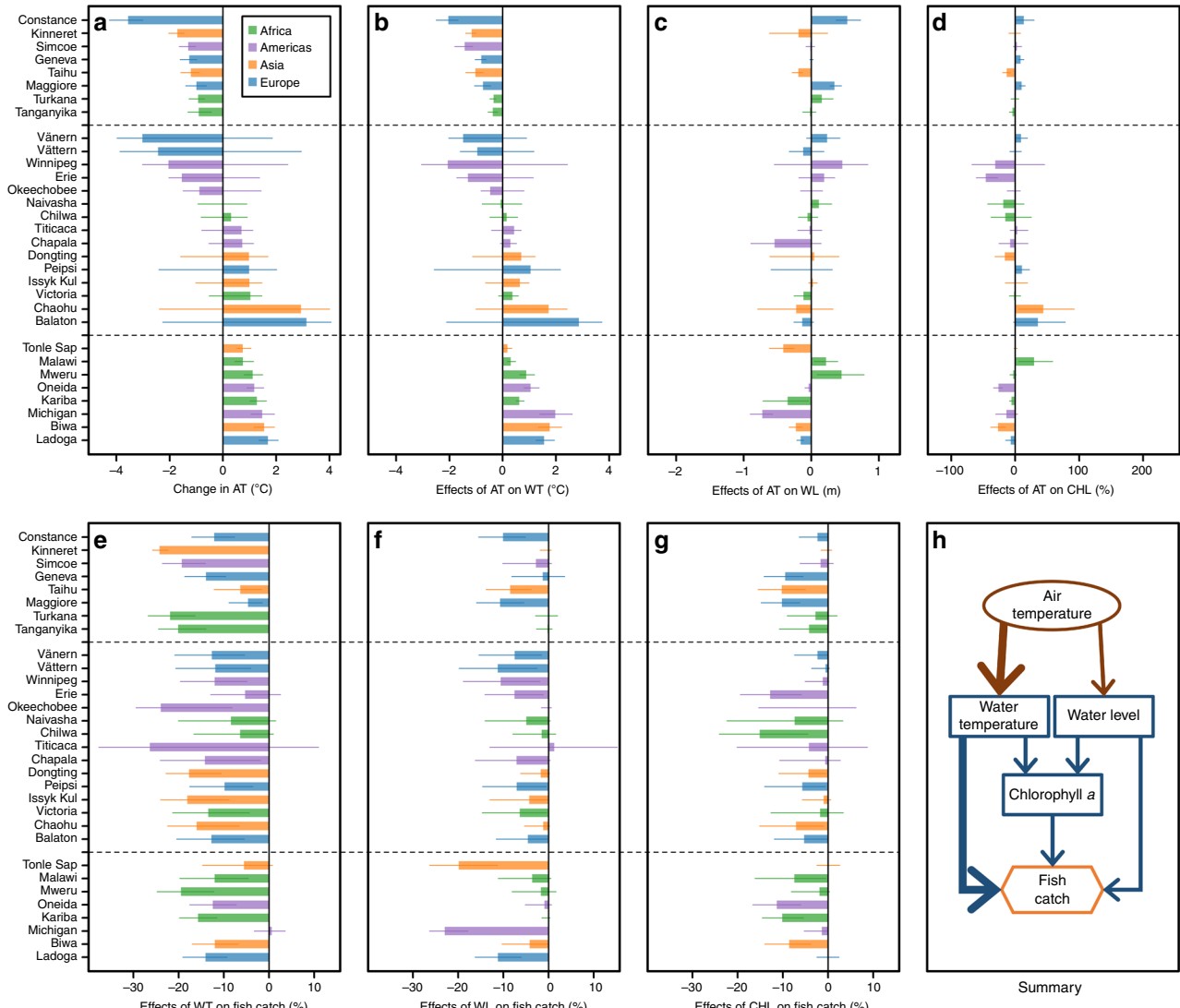

**Fig. 4 Assessment for effects of air temperature on fish catches across the 31 study lakes.** Changes or effects were calculated as differences (actual or relative) from the 1970–2014 medians. **a** shows simulated changes in air temperatures (AT) associated with a 25% decrease in fish catches. **b–d** show simulated effects of changes in AT on water temperature (WT), water level (WL), and chlorophyll a (CHL). **e–g** depict how simulated effects of WT, WL, and CHL, in response to changing AT, affected fish catches. **h** summarizes these linkages within a subset of the Bayesian networks model that demonstrates the effects of AT, where the width of each arrow is a qualitative indicator of the strength of the hypothesized causal effects. In **a–g**, bars represent simulated median changes and error bars represented first and third quartiles of the simulated changes. Dashed lines divide the lakes into "positive" (top 8 lakes), "mixed" (middle 15 lakes), or "negative" (bottom 8 lakes) associations between AT and fish catches, based on one-tailed 75% credible intervals, as indicated by error bars in **a**. For positive lakes, a 25% decrease in fish catches was associated with colder AT. For mixed lakes, a 25% decrease in fish catches was associated with both colder and warmer AT. For negative lakes, a 25% decrease in fish catches was associated with warmer AT. Source data for **a–g** are provided as a Source Data file.

associated with decreasing LUag (i.e., a positive relationship), with median decreases of 3.0–6.9% of catchment area from the observed 1970–2014 medians (Fig. 6a). In turn, these simulated decreases in LUag resulted in lower CHL for eight of these nine lakes, but a higher CHL in the other (Fig. 6c). For another 12 lakes, a simulated 25% catch decrease was associated with increasing LUag (i.e., a negative relationship), with median increases of 1.2–14.0% of catchment area from the observed 1970–2014 medians (Fig. 6a). However, the simulated effects of increasing LUag on CHL were more variable across lakes, some associated with higher CHL and some associated with lower CHL (Fig. 6c). Translating these simulated changes to effects on CATCHs revealed that the effects of CHL, resulting from changes in LUag, were important in all 21 lakes (i.e., leading to a >13.4% catch decrease), but once again the direction of the relationships

varied across lakes, particularly when LUag increased. The effects of WL, resulting from changes in LUag, on catch decrease were unimportant in these 21 lakes. For the remaining 10 lakes, a mixed relationship was revealed as a simulated 25% catch decrease was associated with both lower and higher LUag.

**Characteristics of lakes where CATCHs are vulnerable.** Our results showed that access to clean water (as measured by the proportion of population in the catchment using drinking-water and sanitation services[37]) was positively correlated with the magnitude of changes in each of AT, PRE, and LUag that was associated with a 25% catch decrease across the study lakes (AT: $r = 0.56$, t-test $p < 0.01$, $N = 31$; PRE: $r = 0.51$, t-test $p < 0.01$, $N = 31$; LUag: $r = 0.50$, t-test $p < 0.01$, $N = 31$; Fig. 7a–c). This

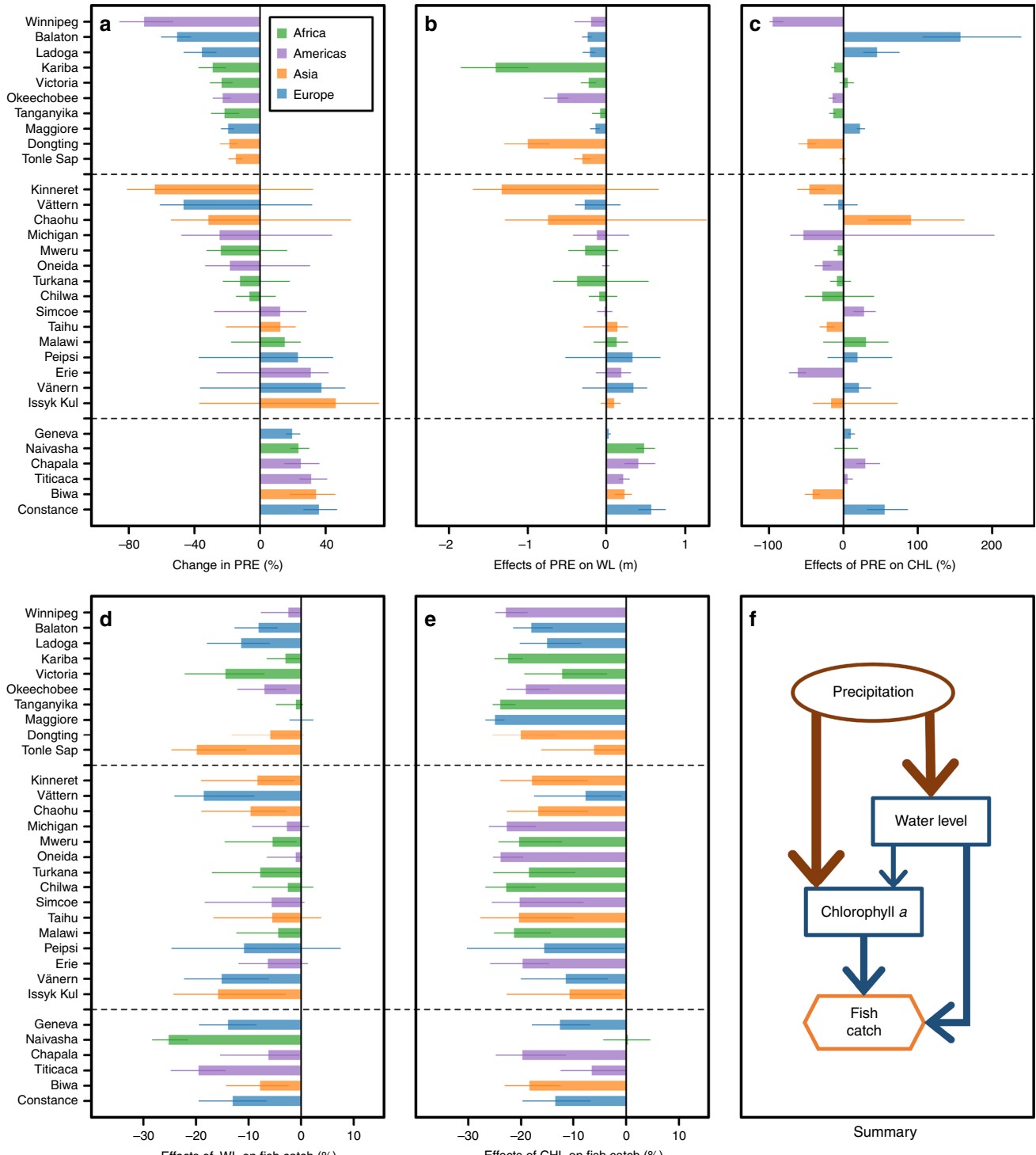

**Fig. 5 Assessment for effects of precipitation on fish catches across the 31 study lakes.** Changes or effects were calculated as differences (actual or relative) from the 1970–2014 medians. **a** shows simulated changes in precipitation (PRE) associated with a 25% decrease in fish catches. **b**, **c** show simulated effects of changes in PRE on water level (WL) and chlorophyll *a* (CHL). **d**, **e** depict how simulated effects of WL and CHL, in response to changing PRE, affected fish catches. **h** summarizes these linkages within a subset of the Bayesian networks model that demonstrates the effects of PRE, where the width of each arrow is a qualitative indicator of the strength of the hypothesized causal effects. In **a–e**, bars represent simulated median changes and error bars represented first and third quartiles of the simulated changes. Dashed lines divide the lakes into "positive" (top 10 lakes), "mixed" (middle 15 lakes), or "negative" (bottom 6 lakes) associations between PRE and fish catches, based on one-tailed 75% credible intervals, as indicated by error bars in **a**. For positive lakes, a 25% decrease in fish catches was associated with lower PRE. For mixed lakes, a 25% decrease in fish catches was associated with both lower and higher PRE. For negative lakes, a 25% decrease in fish catches was associated with higher PRE. Source data for **a–e** are provided as a Source Data file.

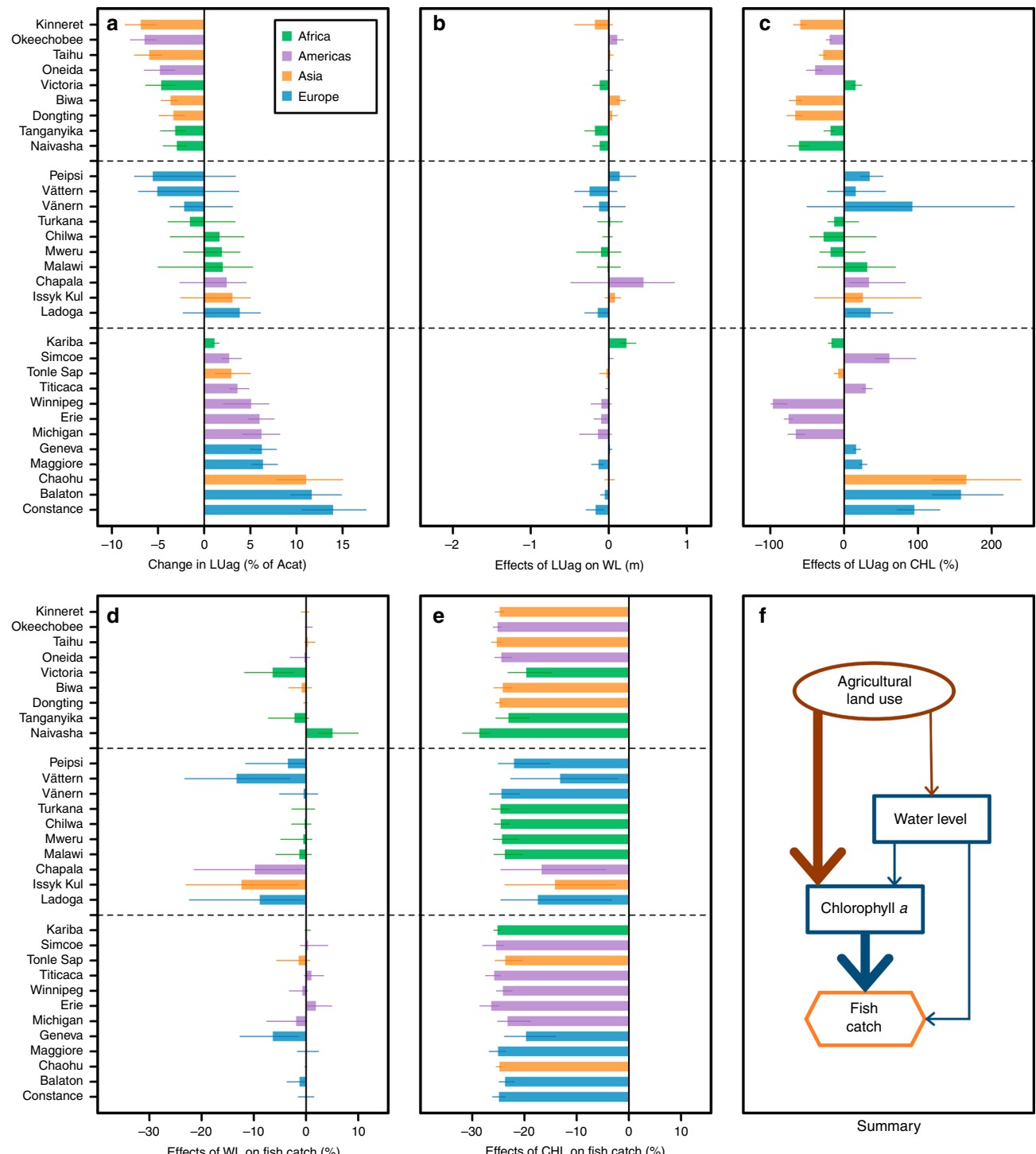

**Fig. 6 Assessment for effects of agricultural land use on fish catches across the 31 study lakes.** Changes or effects were calculated as differences (actual or relative) from the 1970–2014 medians. **a** shows simulated changes in agricultural land use (LUag), in terms of percent of catchment area (Acat), associated with a 25% decrease in fish catches. **b**, **c** show simulated effects of changes in LUag on water level (WL) and chlorophyll *a* (CHL). **d**, **e** depict how simulated effects of WL and CHL, in response to changing LUag, affected fish catches. **h** summarizes these linkages within a subset of the Bayesian networks model that demonstrates the effects of LUag, where the width of each arrow is a qualitative indicator of the strength of the hypothesized causal effects. In **a**–**e**, bars represent simulated median changes and error bars represented first and third quartiles of the simulated changes. Dashed lines divide the lakes into "positive" (top 9 lakes), "mixed" (middle 10 lakes), or "negative" (bottom 12 lakes) associations between LUag and fish catches, based on one-tailed 75% credible intervals, as indicated by error bars in **a**. For positive lakes, a 25% decrease in fish catches was associated with decreasing LUag. For mixed lakes, a 25% decrease in fish catches was associated with both decreasing and increasing LUag. For negative lakes, a 25% decrease in fish catches was associated with increasing LUag. Source data for **a**–**e** are provided as a Source Data file.

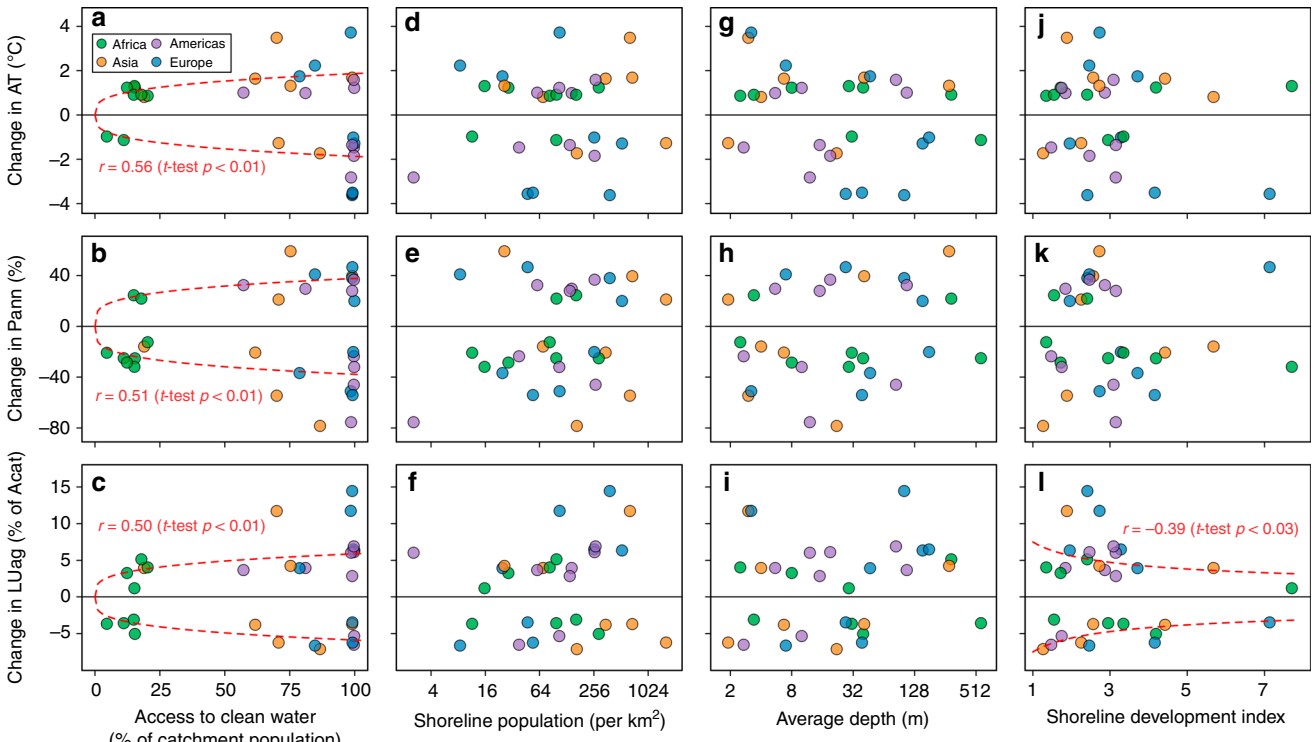

**Fig. 7 Identification of the characteristics of the vulnerable lakes.** The figure shows simulated median magnitudes of changes, from the 1970–2014 medians, for air temperature (AT), precipitation (PRE), and agricultural land use (LUag, in terms of percent of catchment area (Acat)) associated with a 25% decrease, also from the 1970–2014 medians, in fish catches and their associations with access to clean water (**a–c**), shoreline population density (**d–f**), average depth (**g–i**), and shoreline development index (SDI; **j–l**) across the 31 study lakes. Note that points in the negative section have negative simulated median changes associated with a 25% decrease in fish catches. A lake with higher vulnerability of fish catches to climate and land-use changes is indicated by lower magnitudes of changes in AT, PRE, and LUag associated with a 25% decrease, from the 1970–2014 medians, in fish catches. A least-squares line is added when the correlation is significant at $p = 0.05$ level ($t$ test, $N = 31$). The access to clean water is measured by the proportion of population in the catchment using drinking-water and sanitation services. The shoreline population density is the population density within 10 km of a lake's shoreline. SDI is a dimensionless index for circularity of the lake surface and a larger SDI indicates that the lake surface is less circular and may have a larger littoral area relative to lake area[24]. Source data for this figure are provided as a Source Data file.

suggests that limited access to clean water is a characteristic for lakes where CATCHs are vulnerable to both climate and land-use changes, as relatively small changes (either increases or decreases) in AT, PRE, or LUag can cause a 25% catch decrease. Our results also showed that the magnitude of changes in LUag associated with a 25% catch decrease was negatively correlated with the shoreline development index (SDI), a measure of the circularity of lake surface and an indicator of littoral area relative to lake area[24], across the study lakes ($r = -0.39$, $t$-test $p = 0.03$, $N = 31$; Fig. 7l). This suggests that lakes with larger littoral area (as indicated by a higher SDI) were more vulnerable because they required relatively small changes in LUag (either increases or decreases) to cause a 25% catch decrease. Across the study lakes, however, there was no significant correlation (i.e., $t$-test $p > 0.05$, $N = 31$) for all combinations between the magnitude of changes in each of AT, PRE, and LUag associated with a 25% catch decrease and shoreline population density or average depth (Fig. 7d–i), and between the magnitude of changes in each of AT and PRE associated with a 25% catch decrease and SDI (Fig. 7j, k).

## Discussion

By using a Bayesian networks modeling approach to analyze time-series data for 31 lakes across five continents in the period 1970–2014, we found that climate and land-use changes could have both positive and negative effects on CATCHs. At one level, we found that effects of a climate or land-use driver (one of AT, PRE, and LUag in this study) on lake-environmental factors (WT,

WL, and primary productivity) were relatively consistent in directions (e.g., warmer AT always increased WT). However, the effects of a changing lake-environmental factor (e.g., CHL) could have either positive or negative effects on CATCHs across lakes (Table 2).

One likely explanation for the lack of a consistent response across lakes is the abiotic and biotic diversity of lake ecosystems that could not be incorporated within our global model. For example, increases in WT can have both positive and negative effects on fish growth and recruitment[38], but the direction of these effects on individual fish species usually depends upon the thermal tolerance of this species and the thermal environment of the lake[27,39,40]. Within a lake, as most fisheries are multi-species and multi-gear in nature, fishers are able to adapt, usually gradually, to changes in the composition of fish species[6]. There is a rich literature reporting how different fish species responded differently to lake-environmental changes that could be linked to changes in climate and land use[9,41–44], especially in lakes where some species are close to the boundary of their current distributions[40,45]. However, in response to climate and land-use changes, lake environment[21] may change at a very rapid pace so that native species cannot adapt, which, at an extreme, could create conditions for highly productive, introduced/exotic species to flourish and become dominant in CATCHs[11,14,46]. For lakes where the catch and effort could be subdivided into trophic levels or habitat guilds, future research could modify our existing modeling framework to predict catch at relevant trophic levels,

for example, to evaluate whether a consistent catch response across lakes can be detected.

By assessing changes in climate and land-use drivers associated with a substantial, 25%, decrease in CATCHs across the 31 study lakes, we identified key processes that could underlie declines in CATCHs. Specifically, we found that changes in AT influence catches more through changes in WT than through changes in WL or primary productivity (as measured by CHL; Fig. 4h). Our BNM revealed strong positive relationships between AT and WT across all the 31 study lakes, which is consistent with predictions based on energy balance and findings in another global study[21]. Given robust projections for climate warming throughout this century[47], our results suggest that projected increases in AT will result in increases in WT, but will not necessarily lead to widespread declines in CATCHs across lakes, as our simulations revealed that decreases in CATCHs were associated with warmer AT in only 7 of the 31 study lakes. One important caveat is that the largest AT increase in our model was 2.6 °C warmer than the 1970–2014 median. We caution against extrapolating our results should AT warm to even higher temperatures.

We found that changes in PRE could lead to substantial declines in CATCHs through changes in either WL or primary productivity (as measured by CHL; Fig. 5f). Across 29 of the 31 study lakes, our BNM suggested strong positive relationships between PRE and WL, which is consistent with water cycle and runoff dynamics. However, the lack of any consistent pattern of linkages between PRE and primary productivity revealed in our BNM exemplifies the diversity of lake systems. In theory, increased PRE can result in increases in primary productivity in a lake, directly through increasing inputs of allochthonous-derived nutrients from the catchment[24] and indirectly through increasing WL that lead to the release of nutrients from inundated riparian land[48]. Nonetheless, depending on the timing, duration, and intensity of PRE, the increased PRE can also result in increases in sediment loads and resuspension, which can result in decreases in water clarity and primary productivity[18]. Based on current projections for this century, generally, future PRE may increase in the high-latitude (>40°) and near-equator (latitude <10°) regions, but decrease in mid-latitude regions[47]. As shown in Supplementary Data 2, these projections suggest that 21 of the 31 study lakes will have PRE increases, yet the scenarios under which these 21 lakes experienced decreases in CATCHs in our simulations included increased PRE (four lakes), reduced PRE (eight lakes), and mixed (nine lakes). While these projections suggest that 10 of the 31 study lakes will have PRE decreases (Supplementary Data 2), the scenarios under which these 10 lakes experienced decreases in CATCHs in our simulations also included increased PRE (two lakes), reduced PRE (two lakes), and mixed (six lakes). As a result, although projected changes in PRE will result in consistent directional changes in WL, it will not necessarily result in the same directional changes in primary production and CATCHs across lakes. Finally, although the projected changes in PRE across global change scenarios[47] are generally within the range of our BNM inputs (i.e., from 63% lower to 51% higher than the 1970–2014 median), we caution against extrapolating our results should PRE change beyond this range.

While increases in LUag could affect lake ecosystems increasing water withdrawals and nutrient inputs, which could result in decreases in WL and increases in primary productivity[2], we found that changes in LUag affected CATCHs more through changes in primary productivity (as measured by CHL) than through changes in WL (Fig. 6f). However, our BNM showed that increases in LUag could result in increases in primary productivity in 17 of the 31 study lakes, which suggests that in the remaining 14 study lakes, the positive relationship between nutrient inputs and LUag was decoupled. One plausible

explanation for this decoupling is that the intensity of agriculture in a lake's catchment could increase with little or no change in the area of agricultural land, such as through increases in fertilizer use[49], livestock densities[49], or cage culture[50]. Additionally, the area of LUag does not reflect the management efforts to control nutrient enrichment in recipient lakes, which is very common in developed countries[2,43], but less so in developing countries.

Generally, LUag is projected to increase in developing countries but decrease in developed countries in this century[51]. As shown in Supplementary Data 2, 16 of the 31 study lakes are projected to have decreases in LUag and the scenarios under which these lakes experienced decreases in CATCHs in our simulations included increased LUag (eight lakes), reduced LUag (two lakes), and mixed (six lakes). Similarly, among 15 lakes that are projected to undergo increases in LUag, the scenarios under which these lakes experienced decreases in CATCHs in our simulations included increased LUag (four lakes), reduced LUag (seven lakes), and mixed (four lakes). Once again, our results suggest that projected changes in LUag will not necessarily result in the same directional changes in primary production and CATCHs across lakes and we caution against extrapolating our results should agricultural land-use change beyond the range of our BNM inputs (from an decrease of 13% to an increase of 11%, in terms of percent of catchment area, from the 1970–2014 median).

We identified low access to clean water (as measured by the proportion of population using drinking-water and sanitation services in the catchment) as one characteristic of lakes where CATCHs are vulnerable to changes in AT, PRE, and LUag (Figs. 7a–c and 8). Access to clean water can only be improved with substantial investments for water-use efficiency and water-quality protection[37]. Therefore, combining with the identified key processes that underlie changes in CATCHs (Figs. 4h, 5f, and 6f), these results suggest that the investments for water-use efficiency and water-quality protection can also make CATCHs less vulnerable to climate and land-use changes. Future research will be required to determine whether this correlation is causative; one potential mechanism that has support from previous studies is that investments in sanitation improve fish habitat[2,9]. One caveat is that lakes in regions with more access to clean water may also have stronger fishing regulations, which could also make CATCHs less vulnerable to climate and land-use changes by sustaining healthier fish stocks. We were unable to disentangle these effects in our analysis because none of the 31 study lakes has both high access to clean water and limited or weak fishing regulations.

Independent of access to clean water, our results suggest that climate and land-use changes can result in substantial decreases in CATCHs across lakes with very different socio-economic and hydrogeomorphological characteristics, such as shoreline population densities, depths, and littoral areas (Figs. 7d–k and 8). Although we identified a large littoral area (as indicated by SDI) as one characteristic for lakes where CATCHs are vulnerable to changes in LUag, the significant relationship was driven by two lakes with SDI >7 (Fig. 7l). Therefore, more research is needed to confirm this suggestion.

Throughout this study, we interpreted the results across the 31 study lakes collectively, with a focus on understanding effects of climate and land-use on CATCHs at a global scale (Figs. 3–7). We caution against interpreting our lake-specific results because the uncertainty around some predictions can be large, and it is difficult to discern whether this is a consequence of structural errors in our BNM or the uncertainty that arises from using data from so many sources (Supplementary Data 3). However, we suggest using our lake-specific predictions as priors for future lake-specific assessments, where quantitative models can be

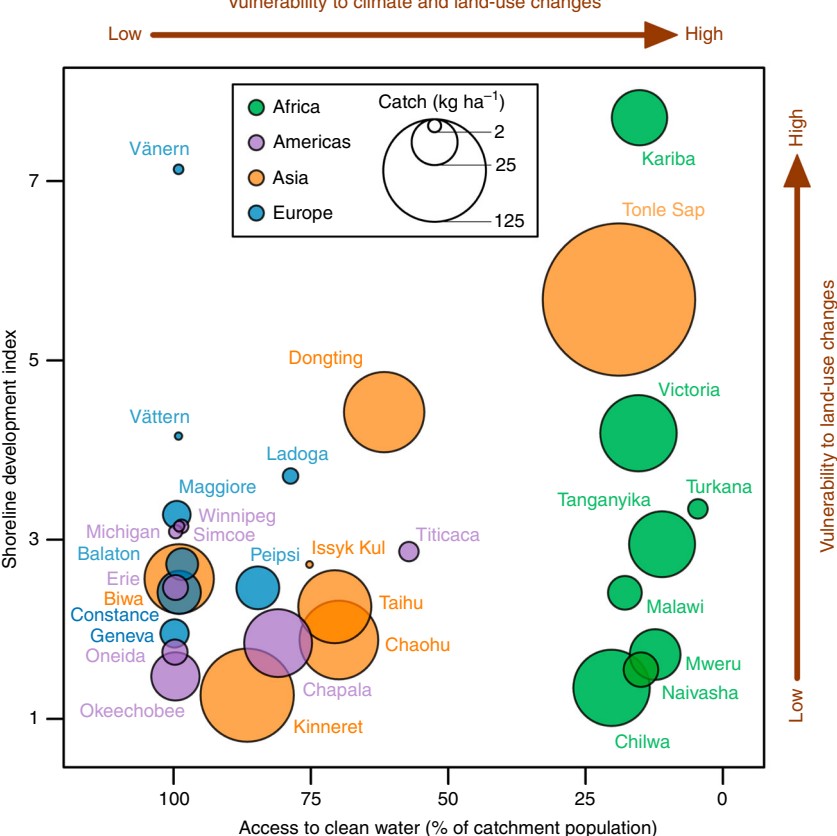

**Fig. 8 Distribution of the 31 study lakes with respect to access to clean water and shoreline development index.** The area of each circle is proportional to the median fish catch in the period 1970–2014. Based on the correlation analysis shown in Fig. 7, lakes located in areas with lower access to clean water are more vulnerable to substantial decreases in fish catches driven by either climate or land-use change and lakes with larger shoreline development index are more vulnerable to substantial decreases in fish catches driven by land-use change. Source data for this figure are provided as a Source Data file.

tailored to include important, lake-specific drivers that we were not able to include in our global model and data collected by consistent methods can be obtained. This is especially relevant to lakes where effects of climate or land-use change alone on CATCHs has not been as prevalent as effects of other anthropogenic stressors, such as pollution, invasive species, and overfishing; however, climate and land-use changes may exacerbate the negative effects of these stressors on CATCHs[39].

In conclusion, although we were unable to identify consistent directional relationships between lake-environmental variables (e.g., WT, WL, CHL) and CATCH (Table 2; Figs. 4–6), correlations based on our simulations suggested an intriguing and potentially important relationship between access to clean water and vulnerability to a large reduction in CATCH in response to climate and land-use changes (Figs. 7 and 8). As an extension, our results suggest that CATCHs for lakes in developing countries in sub-Saharan Africa, Southeastern and Central Asia, and Central and South America are more vulnerable, which is consequential given that inland CATCHs are among the highest in the world in these regions and that the main purpose of fishing is for food rather than income[6]. To mitigate the threat of significant decreases in CATCHs and food security, our results imply that development strategies that include investments for clean water[37] can also benefit lake CATCHs in the face of anthropogenic stressors (Fig. 8). The implications of this result made possible by analyzing time series of lakes across continents highlight possible synergies for policy makers aiming to achieve multiple United Nation Sustainable Development Goals (SDGs)[52], including Zero Hunger (SDG 2), Clean Water and Sanitation (SDG 6), Climate

Action (SDG 13), and Life Below Water (SDG 14), and Life on Land (SDG 15).

## Methods

**Choice of modeling approach**. We chose to use a Bayesian networks modeling approach because it could minimize negative effects of data limitations on the statistical power of our analysis, by accounting for our understanding of multilevel, causal processes that could underlie changes in CATCHs in response to changes in climate and land use across lakes. A BNM could be developed to represent the hypothesized causal processes by expressing them as conditional probabilistic relationships between model variables, with which the estimation of every BNM coefficient took all model inputs derived from data into account[20]. For example, if we developed a simple BNM to represent that changes in AT resulted in changes in WT, which, in turn, resulted in changes in CATCH in a lake, the estimation of model coefficients for the relationship between AT and WT and for the relationship between WT and CATCH would both take model inputs for all variables (AT, WT, and CATCH) into account. Although the 31 study lakes were selected partly because of good data availability, there were still large within-lake differences in data availability among model variables (as shown in Supplementary Data 3). For example, we obtained 9, 16, and at least 32 years of data for CHL, WT, and all the other variables in Lake Chilwa, respectively. Therefore, the BNM's capability of taking all model inputs into account for the estimation of every model coefficient was important to maximize the statistical power of our analysis.

**Model development**. To mathematically formulate our BNM based on our hypothesized causal processes (Fig. 2), we assumed that (1) each response variable in the BNM has a normal distribution, with a mean that can be expressed as a function of predictor variables based on the primary literature described below, and (2) annual time-series data are independent observations, such that the mean value of each response variable in 1 year is dependent upon the values of predictor variables in the same year, but independent from all of its mean values in the other years. For example, the conditional probability distribution of WT in our

BNM was expressed as

$$\begin{cases} \text{WT}_i \sim \text{NORM}(\mu_i^{\text{WT}}, \sigma^{\text{WT}}), \\ \mu_i^{\text{WT}} = \beta_{0,i}^{\text{WT}} + \beta_{1,i}^{\text{WT}} \times \text{AT}_i, \end{cases} \quad (1)$$

where $\text{NORM}(\mu, \sigma)$ represents a normal distribution with a mean of $\mu$ and a standard deviation of $\sigma$, subscripted $i$ indicates the $i$th lake, superscripted WT indicates that the parameter or BNM coefficient was associated with the response variable WT, and $\beta$ represents a BNM coefficient. In Eq. (1), $\mu^{\text{WT}}$ was expressed as a linear function of AT based on the primary literature[21] and was assumed to be dependent upon the observed value of AT in the same year, but independent from any $\mu^{\text{WT}}$ of the other years. In a similar manner, we give details for how we developed the conditional probability distributions of the other three BNM response variables—change in water level ($\Delta$WL), CHL, and CATCH—in the following of this section.

The conditional probability distribution of $\Delta$WL in our BNM was developed based on an empirical linear relationship between $\Delta$WL and annual catchment runoff (RUNOFF)[23], of which the latter could be expressed as a balance between PRE and evapotranspiration (ET)[53]. In a simple manner, these relationships could be expressed as

$$\Delta\text{WL} = \alpha_0^{\Delta\text{WL}} + \alpha_1^{\Delta\text{WL}} \times \text{RUNOFF}, \quad (2)$$

$$\text{RUNOFF} = \text{PRE} - \alpha_1^{\text{ET}} \times \text{PE} \times \text{LUag} - \alpha_2^{\text{ET}} \times \text{PE} \times \text{LUun}, \quad (3)$$

where $\alpha$ represents an empirical coefficient, PE is potential evaporation, LUun is undeveloped land use, and $\alpha_1^{\text{ET}} \times \text{PE} \times \text{LUag}$ and $\alpha_2^{\text{ET}} \times \text{PE} \times \text{LUun}$ represent ET over agricultural area and undeveloped area, respectively. In this study, we categorized catchment land use into three types: agricultural (which includes cropland, managed pasture, and rangeland), undeveloped (which includes forest, primary and secondary non-forest land, the lake itself, and other water and ice area), and urban. Equation (3) did not include urban land use because urban water use is mostly non-consumptive and returns to the catchment[54]. By combining Eqs. (2) and (3) and re-arranging $\alpha$ coefficients, the conditional probability distribution of $\Delta$WL in our BNM was expressed as

$$\begin{cases} \Delta\text{WL}_i \sim \text{NORM}(\mu_i^{\Delta\text{WL}}, \sigma^{\Delta\text{WL}}), \\ \mu_i^{\Delta\text{WL}} = \beta_{0,i}^{\Delta\text{WL}} + \beta_{1,i}^{\Delta\text{WL}} \times \text{PRE}_i + \beta_{2,i}^{\Delta\text{WL}} \times \text{PE}_i + \beta_{3,i}^{\Delta\text{WL}} \times \text{PE}_i \times \text{LUag}_i, \end{cases} \quad (4)$$

where each $\beta$ coefficient can be expressed as a combination of $\alpha$ coefficients in Eqs. (2) and (3). We only included LUag in Eq. (4) because in each of these 31 study lakes, LUag + LUun in the catchment was almost a constant, which means that LUun could be replaced with a constant minus LUag, as the increase in agricultural area mostly resulted from the decrease in undeveloped area, or vice versa, in the study period.

The conditional probability distribution of CHL in our BNM was developed based on empirical log–log relationships between CHL and total phosphorus[24] and between CHL and WT[25]. Because total phosphorus data were only available for a few of our study lakes, we modeled the total phosphorus in lakes as a function of PRE[26], LUag[26], and $\Delta$WL[29]. This led to a conditional probabilistic distribution for CHL expressed as

$$\begin{cases} \log(\text{CHL}_i) \sim \text{NORM}(\mu_i^{\text{CHL}}, \sigma^{\text{CHL}}), \\ \mu_i^{\text{CHL}} = \beta_{0,i}^{\text{CHL}} + \beta_{1,i}^{\text{CHL}} \times \log(\text{PRE}_i) + \beta_{2,i}^{\text{CHL}} \times \log(\text{LUag}_i) + \beta_{3,i}^{\text{CHL}} \times \mu_i^{\Delta\text{WL}} + \beta_{4,i}^{\text{CHL}} \times \mu_i^{\text{WT}}, \end{cases} \quad (5)$$

where $\mu^{\text{WT}}$ and $\mu^{\Delta\text{WL}}$ are from Eqs. (1) and (4), respectively.

Finally, the conditional probability distribution of CATCH was developed based on (1) a theoretical relationship between CATCH and fish biomass ($B$)[34], (2) a hypothesized empirical relationship between fishing catchability ($Q$) and EFF and $B$[34], and (3) a hypothesized empirical relationship between $B$ and WT[27,28], $\Delta$WL[29,30], and CHL[31,32] and number of fish stocked (ST)[35,36]. In a simple manner, these relationships could be expressed as

$$\text{CATCH} = Q \times \text{EFF} \times B \Rightarrow \log(\text{CATCH}) = \log(Q) + \log(\text{EFF}) + \log(B), \quad (6)$$

$$\log(Q) = \alpha_0^Q + \alpha_1^Q \times \log(\text{EFF}) + \alpha_2^Q \times \log(B), \quad (7)$$

$$\log(B) = \alpha_0^B + \alpha_1^B \times \text{WT} + \alpha_2^B \times \Delta\text{WL} + \alpha_3^B \times \log(\text{CHL}) + \alpha_4^B \times \log(\text{ST}), \quad (8)$$

By combining Eqs. (6)–(8) and re-arranging $\alpha$ coefficients, we developed the conditional probabilistic distribution for CATCH expressed as

$$\begin{cases} \log(\text{CATCH}_i) \sim \text{NORM}(\mu_i^{\text{CATCH}}, \sigma^{\text{CATCH}}), \\ \mu_i^{\text{CATCH}} = \beta_{0,i}^{\text{CATCH}} + \beta_{1,i}^{\text{CATCH}} \times \mu_i^{\text{WT}} + \beta_{2,i}^{\text{CATCH}} \times \mu_i^{\Delta\text{WL}} + \beta_{3,i}^{\text{CATCH}} \times \mu_i^{\text{CHL}} \\ \quad\quad + \beta_{4,i}^{\text{CATCH}} \times \log(\text{ST}_i) + \beta_{5,i}^{\text{CATCH}} \times \log(\text{EFF}_i), \end{cases} \quad (9)$$

where $\mu^{\text{WT}}$, $\mu^{\Delta\text{WL}}$, and $\mu^{\text{CHL}}$ are from Eqs. (1), (4), and (5), respectively. As described in the Supplementary Methods, we did not estimate $\beta_{4,i}^{\text{CATCH}}$ for 23 lakes where stocking effects on CATCHs were not sufficiently important in the study period.

The derivations and data sources of model inputs, including climate and land-use drivers (AT, PRE, and LUag), lake-environmental factors (WT, WL, and CHL), CATCH, ST, and EFF are given in the Supplementary Methods.

**Prior distributions.** We followed a previously used procedure[20] to derive prior distributions of BNM coefficients. For example, to derive the prior distribution of $\beta^{\text{WT}}$, we first expressed the function of $\mu_i^{\text{WT}}$ in Eq. (1) as 31 linear models (i.e., one for each study lake) in a form

$$\text{WT}_i = \beta_{0,i}^{\text{WT}} + \beta_{1,i}^{\text{WT}} \times \text{AT}_i + \varepsilon_i^{\text{WT}}, \quad (10)$$

where $\varepsilon$ is the residual error. Then we derive the least-squares estimates for the coefficients of each of these 31 linear models based on model inputs derived from data for $\text{WT}_i$ and $\text{AT}_i$ in the same year. In theory, the least-squares estimates for the coefficients of each of these 31 linear models have a multivariate normal (MVN) distribution because they could be expressed as linear combinations of $\text{WT}_i$, which was assumed to have a normal distribution as shown in Eq. (1). Finally, we used the MVN distribution for the least-squares estimates of the linear model coefficients as the prior distribution of each of the 31 sets of $\beta_i^{\text{WT}}$, but the covariance matrix of the MVN distribution was set to be proportional to its un-scaled, least-squares estimates, with a proportional constant (i.e., the hyperparameter) having a noninformative (hyperprior) distribution[20]. We repeated this procedure to derive prior distributions for $\beta_i^{\Delta\text{WL}}$, $\beta_i^{\text{CHL}}$, and $\beta_i^{\text{CATCH}}$. The prior distributions of these BNM coefficients were always developed on a lake-by-lake basis because for each of the 31 study lakes, we were able to obtain data to derive model inputs for at least 9 years for all BNM variables (refer to Supplementary Data 3 for data sources and availability). Lastly, for nuisance parameters $\sigma^{\text{WT}}$, $\sigma^{\Delta\text{WL}}$, $\sigma^{\text{CHL}}$, and $\sigma^{\text{CATCH}}$ in Eqs. (1), (4), (5), and (9), respectively, we set each of them to have a noninformative prior distribution.

**Coefficient estimation.** The posterior distributions of our BNM coefficients were developed based on pooled information across the 31 study lakes. Our BNM was structured in a way to have the transferability of information across the lakes informed by data. Specifically, in the derivation of posterior distributions of our BNM coefficients, the relative importance of lake-specific information increases with lake-specific sample size and across-lake difference. If a lake was very different from all the other lakes, for example, the relative importance of its lake-specific information would be much higher than across-lake information in the derivation of posterior distributions of lake-specific BNM coefficients.

Computationally, we derived empirical posterior distributions for all BNM coefficients by using a Gibbs sampling method to run Markov chain Monte Carlo (MCMC) simulations[55]. We ran three random-starting MCMC chains by implementing the program JAGS[56] and the package runjags[57] in R[58]. After discarding 20,000 iterations for burn-in and adaptation, we considered that the convergence of three MCMC chains was achieved at 50,000 interactions as autocorrelations were close to 0 and Gelman and Rubin's convergence diagnostics[59] were <1.005 for all coefficients. Our empirical posterior distribution of each BNM coefficient was based on 50,000 samples from each converged MCMC chain.

**Effects between variable pairs.** As shown in Fig. 2, there were 13 pairs of predictor and response variables in our BNM. For each of these 13 variable pairs, we categorized the effects of predictor variable on response variable as positive, negative, and mixed in each lake based on the posterior distribution of the associated BNM coefficient, which we used as a proxy of effect strength because it represents how much the response variable would change in response to one unit change in the predictor variable. For example, we categorized the effects of AT on WT in lake $i$ based on the posterior distribution of $\beta_{1,i}^{\text{WT}}$, which represents how much of WT would increase in response to one unit change of AT. We categorized effects between a variable pair as positive and negative based on the 75% CI of the associated BNM coefficient. We categorized effects between a variable pair as mixed when both positive and negative 75% CIs of the associated BNM coefficients included 0, which, statistically, also means that its 50% CI included 0.

**Changes in drivers associated with a 25% catch decrease.** We implemented the BNM and used a Monte Carlo simulation method to empirically derive distributions of climate (AT and PRE) and land-use (LUag) drivers associated with a 25% decrease in CATCH from its 1970–2014 median (i.e., a 25% catch decrease) in each of the 31 study lakes. We used a 25% catch decrease as our simulation target because it is close to the maximum value of standard errors for the ratios of CATCH to its median, which ranged from 2.1% to 25.7% across the study lakes in the study period.

**Monte Carlo simulation.** Our Monte Carlo simulation was a three-step process. The first step was to randomly sample a set of BNM coefficients (i.e., $\hat{\beta}^{\text{WT}}$, $\hat{\beta}^{\Delta\text{WL}}$, $\hat{\beta}^{\text{CHL}}$, and $\hat{\beta}^{\text{CATCH}}$) from empirical posterior distributions by implementing the package rv[60] in R. Then, we calculated predicted medians of WT, $\Delta$WL, CHL, and

CATCH (denoted as $\mu_0^{WT}$, $\mu_0^{\Delta WL}$, $\mu_0^{CHL}$, and $\mu_0^{CATCH}$, respectively) using the set of BNM coefficients and the 1970–2014 medians of all BNM inputs.

The second step was to generate a set of BNM inputs from probability distributions based on lake-by-lake data by implementing random number generators in R. We assumed that each of AT, PRE, and LUag has a normal distribution. PE was derived empirically based on randomly sampled AT and the assumption that temperature change is uniform throughout the year (e.g., if randomly sampled AT was 1 °C higher than its 1970–2014 median, we assumed that AT in every month was 1 °C higher than its 1970–2014 median).

The third step was to calculate predicted values (i.e., $\mu^{WT}$, $\mu^{\Delta WL}$, $\mu^{CHL}$, and $\mu^{CATCH}$) using randomly sampled BNM coefficients, randomly sampled values of one climate or land-use drivers, and 1970–2014 medians of the other climate and/or land-use variables not being evaluated. For example, to investigate effects of AT on CATCH, we calculated BNM predicted values using BNM coefficients randomly sampled from empirical posterior distributions, inputs of AT randomly sampled from normal distributions, inputs of PE derived based on random samples of AT, and 1970–2014 medians of PRE and LUag.

We kept the set of BNM coefficients, inputs, and predicted values that led to a ratio of predicted CATCH to predicted median CATCH, which was calculated as $[\exp(\mu^{CATCH} - \mu_0^{CATCH})]$, between 0.74 and 0.76, and assumed they were associated with a 25% catch decrease. We repeated the Monte Carlo simulation process until we obtained 10,000 samples associated with a 25% catch decrease for each lake to derive empirical probability distributions.

**Assessment of climate and land-use effects**. We assessed how climate and land-use drivers (i.e., AT, PRE, and LUag) could lead to a 25% catch decrease by analyzing empirical probability distributions derived from samples obtained in our Monte Carlo simulations.

To assess effects of AT on CATCH in each of the 31 study lakes, we analyzed empirical probability distributions associated with a 25% catch decrease for changes in AT, predicted effects of AT on WT, WL, and CHL, and predicted effects of WT, WL, and CHL on CATCH. We used predicted changes in WT, WL, and CHL driven by AT to quantify effects of AT on WT, WL, and CHL, respectively. Following the methods used by the Intergovernmental Panel on Climate Change (IPCC)[47], we calculated predicted changes as an actual difference from the 1970–2014 medians for AT, WT, and WL and as a percent difference from the 1970–2014 median for CHL. We used predicted changes in CATCH driven by WT, WL, and CHL to quantify effects of WT, WL, and CHL on CATCH, respectively. The predicted changes in CATCH driven by WT, WL, and CHL were calculated as percent differences from the 1970–2014 medians based on Eq. (9). For example, we calculated predicted effects of WT on CATCH as

$$\{\exp[\beta_1^{CATCH} \times (\mu^{WT} - \mu_0^{WT})] - 1\} \times 100\%. \tag{11}$$

Similar to how we summarized BNM coefficients, we also categorized relationships between AT and CATCH and between AT and each of WT, WL, and CHL as positive, negative, and mixed based on 75% CIs of their changes associated with a 25% catch decrease in each lake. Further, we considered effects of WT, WL, or CHL on CATCH to be important in each lake if its predicted effects could lead to a >9.1% catch decrease based on 75% CI. The cutoff value 9.1% was calculated as $(1-0.75^{1/3}) \times 100\%$, based on the assumption that WT, WL, and CHL contributed equally to a 25% catch decrease.

With two changes, we followed the same method used to assess AT effects on CATCH to assess effects of PRE and effects of LUag on CATCH. First, following the IPCC methods for calculating projected global distributions of changes in PRE and land use, we calculated predicted changes as a percent difference from the 1970–2014 median for PRE and as an actual difference from the 1970–2014 median for LUag. Second, we did not analyze changes in WT and effects of WT on CATCH in both assessments, because there was no hypothesized linkage between PRE and WT and between LUag and WT (Fig. 2). Consequently, since effects of WT on CATCH were not assessed, we considered effects of WL or CHL on CATCH to be important in each lake if its predicted effects could lead to a >13.4% catch decrease based on 75% CI. The cutoff value 13.4% was calculated as $(1-0.75^{1/2}) \times 100\%$, based on the assumption that WL and CHL contributed equally to a 25% catch decrease.

**Characteristics of lakes where CATCHs are vulnerable**. We define vulnerability of a lake as the extent to which the focal unit (e.g., CATCH) will be adversely affected by environmental changes[61]. In this study, we used the magnitude of change for each of AT, PRE, and LUag associated with a 25% catch decrease to index the vulnerability of a lake to a 25% catch decrease owing to climate and land-use changes. For example, we considered a hypothetical lake A to be more vulnerable than another hypothetical lake B if the magnitude of change for each of AT, WL, and CHL associated with a 25% catch decrease was smaller in lake A than lake B.

We investigated whether the vulnerability of a lake to a 25% catch decrease corresponded to any of the two socio-economic characteristics associated with the catchment: access to clean water and shoreline population density; and any of the two hydrogeomorphological characteristics of the lake: average depth and SDI. The access to clean water was indicated by the proportion of population using drinking-

water and sanitation services in the catchment[37]. The shoreline population density was the population density within 10 km of a lake's shoreline. SDI is a measure of circularity of the lake surface[24], which is defined as

$$SDI = 0.5 \times SL/(\pi \times Alake)^{0.5}, \tag{12}$$

where SL is shoreline length and Alake is lake area. A larger SDI indicates that the lake surface is less circular and may have a larger littoral area relative to lake area. The derivations and values of these socio-economic and hydrogeomorphological characteristics are given in the Supplementary Methods.

We conducted a correlation analysis to evaluate whether the relationship between each of the 4 socio-economic and hydrogeomorphological characteristics and the magnitude of change for each of AT, PRE, and LUag associated with a 25% catch decrease was significant at $p = 0.05$ level (t test, $N = 31$). To quantify the magnitude of change for each of AT, WL, and CHL associated with a 25% catch decrease, we used the predicted median of absolute change associated with a 25% catch decrease, which was calculated from empirical probability distributions derived from samples obtained in our Monte Carlo simulations.

**Reporting summary**. Further information on research design is available in the Nature Research Reporting Summary linked to this article.

## Data availability
Data used to derive model inputs for variables associated with lake environment, fish catch, fish stocking, and fishing effort (i.e., WT, ΔWL, CHL, CATCH, ST, and EFF) are either publicly available or available upon requests to corresponding authorities, as given in Supplementary Data 3. Data used to derive model inputs for variables associated with climate and land use (i.e., AT, PRE, PE, and LUag) are from publicly available global databases, as described the Supplementary Methods. The source data underlying Figs. 4a–g, 5a–e, 6a–e, 7, and 8 are provided as a Source Data file.

## Code availability
JAGS code for running MCMC simulations is provided as Supplementary Data 5. R code associated with the estimation of BNM coefficients is provided as Supplementary Data 6. R codes associated with BNM input derivation and BNM simulations are available upon request to the corresponding author.

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

## Acknowledgements

This research was funded by the US Geological Survey National Climate Adaptation Science Center. We thank help from A. Sakas for data compilation and email communication at the beginning of this project. We thank E. Politi for global database support and C. Paukert for comments on an earlier draft of this manuscript. Personal funding supports, data providers, and data sources are acknowledged in Supplementary Note. Any use of trade, firm, or product names is for descriptive purposes only and does not imply endorsement by the US Government.

## Author contributions

Y.-C.K led the work, carried out the study with help from M.W.R., D.B.B., and I.G.C., analyzed data with help from S.S.Q., M.W.R., D.B.B., and I.G.C., and wrote the initial draft of this manuscript with help from M.W.R. and D.B.B. The study was initiated by T.D.B. and initially designed by M.W.R. and D.B.B. O.A., T.D.B., A.B., J.R.B., R.C.-C., N.J.G, J.R.J., K.K., J.K., A.A.L., A.J.L., N.M.-S., R.M.-E., F.J.N., I.O., L.G.R., A.L.E.S., Y.S., H.S.-M., A.T., P.A.M.Z., P.V., Y.W., A.W., O.L.F.W., and J.D.Y. contribute to obtain and compile data used in this study and provided inputs to the analyses and manuscript. Sixth through thirty-first authors contributed equally to this study and are listed alphabetically.

## Competing interests

The authors declare no competing interests.

## Additional information

Yu-Chun Kao[1]✉, Mark W. Rogers[2], David B. Bunnell[3], Ian G. Cowx[4], Song S. Qian[5], Orlane Anneville[6], T. Douglas Beard Jr.[7], Alexander Brinker[8], J. Robert Britton[9], René Chura-Cruz[10], Natasha J. Gownaris[11], James R. Jackson[12], Külli Kangur[13], Jeppe Kolding[14], Anatoly A. Lukin[15], Abigail J. Lynch[7], Norman Mercado-Silva[16], Rodrigo Moncayo-Estrada[17], Friday J. Njaya[18], Ilia Ostrovsky[19], Lars G. Rudstam[12], Alfred L.E. Sandström[20], Yuichi Sato[21], Humberto Siguayro-Mamani[10], Andy Thorpe[22], Paul A.M. van Zwieten[23], Pietro Volta[24], Yuyu Wang[25], András Weiperth[26], Olaf L.F. Weyl[27] & Joelle D. Young[28]

[1]Center for Systems Integration and Sustainability, Department of Fisheries and Wildlife, Michigan State University, 1405 South Harrison Road, East Lansing, MI 48823, USA. [2]US Geological Survey, Tennessee Cooperative Fishery Research Unit, Tennessee Technological University, Box 5114, Cookeville, TN 38505, USA. [3]US Geological Survey, Great Lakes Science Center, 1451 Green Road, Ann Arbor, MI 48105, USA. [4]Hull International Fisheries Institute, University of Hull, Hull HU6 7RX, UK. [5]Department of Environmental Sciences, University of Toledo, Mail Stop 604, Toledo, OH 43606, USA. [6]Centre Alpin de Recherche sur les Réseaux Trophiques des Ecosystèmes Limniques (CARRTEL), Université Savoie Mont Blanc-INRAE, 75 bis avenue de Corzent, 74200 Thonon-les-Bains, France. [7]US Geological Survey, National Climate Adaptation Science Center, 12201 Sunrise Valley Drive, Mail Stop 516, Reston, VA 20192, USA. [8]Fisheries Research Station of Baden-Württemberg, Argenweg 50/1, 88085 Langenargen, Germany. [9]Department of Life and Environmental Sciences, Faculty of Science and Technology, Bournemouth University, Fern Barrow, Poole, Dorset BH12 5BB, UK. [10]Laboratorio Continental de Puno, Instituto del Mar del Perú, Avenida Circunvalación Sur 1911, Barrio San Martin, Puno, Perú. [11]Environmental Studies Department, Gettysburg College, Gettysburg, PA 17325, USA. [12]Cornell Biological Field Station and Department of Natural Resources, Cornell University, 900 Shackelton Point Road, Bridgeport, NY 13030, USA. [13]Centre for Limnology, Institute of Agricultural and Environmental Sciences, Estonian University of Life Sciences, 51117 Rannu, Tartu County, Estonia. [14]Department of Biological Sciences, University of Bergen, P.O. Box 7803, N-5020 Bergen, Norway. [15]Federal Selection and Genetic Centre for Fish Breeding, Federal Agency on Agriculture, Ministry of Agriculture of Russia, Strelninskoe Av., 1, Saint-Petersburg region, Ropsha, Russian Federation. [16]Centro de Investigación en Biodiversidad y Conservación, Universidad Autónoma del Estado de Morelos, Av. Universidad 1001, Col. Chamilpa, C.P. 62209 Cuernavaca, Morelos, México. [17]Instituto Politécnico Nacional-CICIMAR and COFAA, Col. Playa Palo de Santa Rita, Código, Postal 23096 La Paz, B.C.S., México. [18]Malawi Department of Fisheries, P.O. Box 593, Lilongwe, Malawi. [19]Israel Oceanographic and Limnological Research, Kinneret Limnological Laboratory, P.O. Box 447, Migdal 1495001, Israel. [20]Department of Aquatic Resources, Swedish University of Agricultural Sciences, Stångholmsvägen 2, SE-17893 Drottningholm, Sweden. [21]Lake Biwa Environmental Research Institute, 5-34 Yanagasaki, Otsu, Shiga 520-0022, Japan. [22]Faculty of Business and Law, University of Portsmouth, Richmond Building, Portland Street, Portsmouth P01 3DE, UK. [23]Aquaculture and Fisheries Group, Wageningen University, P.O. Box 338, 6700AH Wageningen, The Netherlands. [24]CNR Water Research Institute, Largo Tonolli 50, 28922 Verbania Pallanza, Italy. [25]School of Nature Conservation, Beijing Forestry University, Box 159, Beijing 10083, People's Republic of China. [26]Faculty of Agriculture and Environmental Sciences, Institute of Aquaculture and Environmental Safety, Department of Aquaculture, Szent István University, Páter Károly utca 1, H-2100 Gödöllő, Hungary. [27]DSI/NRF Research Chair in Inland Fisheries and Freshwater Ecology, South African Institute for Aquatic Biodiversity, Makhanda 6140, South Africa. [28]Ontario Ministry of the Environment, Conservation and Parks, 125 Resources Road, Toronto, ON M9P 3V6, Canada. ✉email: kaoyc@msu.edu

