## [Peer Review File · Nature Communications]

Reviewers' comments:

Reviewer #1 (Remarks to the Author):

The authors present a very interesting analysis assessing the interactions between recent climate change, intensification of land use and recorded catch from 31 major freshwater lakes situated across 5 continents. The scope and scale of this study are remarkable and clearly fit within the requirements for this journal. The methods are robust, appropriate for this study and to the best of my knowledge conducted in the correct manner. However, the results, or at least the current presentation of them, are ambiguous and fail to offer a single mechanism describe the environmental drivers and recorded catch. In addition, the paper seems quite long for this journal and I don't feel that 8 large figures are warranted in the main body of the paper. I outline these thoughts in more detail below. Ultimately, I think that while the authors have a unique dataset with which to examine biotic-abiotic relationship of great global importance restructuring of the paper is necessary to allow readers to better interpret their findings.

Bayesian network models are an extremely powerful tool, well suited to the type of study being performed here. I am not an expert in this method however and my lack in comments regarding the details of the authors methodology should not be interpreted as support for it, I simply do not have the knowledge to adequately critique it and defer to other more experienced reviewers in this regard.

Repeatedly, in the abstract and discussion the authors refer to positive and negative relationships observed between the same factors across the study, but do not provide a clear understanding of why causes this variation in responses. While this may be an accurate reflection of their findings it does not advance the general understanding of these relationships. There must be some factor which accounts for why this relationship is positive or negative. Given the availability of data regarding these lakes, I would hope that this factor could be identified and included in the models. Conceivably the type of species (pelagic/ benthic; primary consumer/predator; large/small) or the principal methods of fishery in each lake could play a role here. For example, the authors reference several possible factors in the opening paragraphs of the discussion but these are not included in the model. Considering the flexibility of their Bayesian approach this seems like a limitation to the paper. In any case, I think the authors need to find a way to address this issue for the paper to be suitable for publication in Nature Communications. They have performed a robust analysis of a unique dataset to address a question of ecological, economic importance but the take home message fails to deliver on this opportunity. Should the authors be able to address this, which I think they will, then the paper will be an outstanding contribution to the field and a suitable publication for this journal

Reviewer #3 (Remarks to the Author):

Summary:

The authors report a global analysis of the effects of climate and land use changes on fisheries catches via their effects on lake environments. Their key findings suggest that climate and land use have negative impacts on catch (i.e. decrease in catch), but as a result of changes to the lake environment that occur in either direction (e.g. warmer or colder air temperatures, more or less precipitation, higher or lower chlorophyll-a). Further, the authors find that a reduction in catch is less likely to occur in cleaner lakes, thus highlighting the need to increase water quality in developing countries to simultaneously benefit drinking water and fisheries health under future change.

Overall, the study was interesting and analyzes an impressive data set. The findings seem to be novel and should be of wide interest. The authors use their analysis to draw very general, but important conclusions that link the effects of climate change on lake ecosystems to socio-economic issues. I also appreciated the transparency of the study, as the authors provided all data and code used for the analyses. Below are some comments that hopefully the authors find helpful and could help strengthen the manuscript.

General comments:

- From reading the abstract, I found that one of the most powerful claims the authors make is how water cleanliness can affect how vulnerable a fishery is to changes to climate/land-use. However, it appears that clean lakes are more common in developed countries which would also have more policies to prevent over-fishing. How does the analysis decouple higher drinking water quality from higher fishing regulations in more developed countries, which could also buffer these systems from significant decreases in catch?
- Is catch calculated as a community estimate? If so, is there a size limit for each fishery? Since more developed countries tend to be more discriminate in their fishing, the authors could be more specific on the size ranges that were used to determine catchability coefficients, and thus to estimate total catch for each site. How, for example, did the authors account for lakes in North America and Europe potentially having multiple, yet targeted, fisheries in each lake versus non-targeted or lack of length regulations in African or Asian fisheries?
- Because the sites are located across continents and some tend to be more clustered than others, I wonder if the analysis would benefit from testing the effects of spatial autocorrelation. I'd imagine that over time, sites within close geographical range would have similar climates and land use, so I wonder how reasonable it is to assume that they are independent when trying to unpack the climate versus land use versus socio-economic impacts on fisheries catch. For example, each region might

have very different end-points of in terms of current or future changes to each class of variables tested (e.g. the least clean lakes in developed countries might be cleaner than the cleanest of those in developing countries, or differences in seasonality, photoperiod, etc.).

- Not being an expert with the BNM approach used, I think the manuscript would benefit from a brief mention of what the BNM is, what it does, and why it was chosen above a more simple method of performing a multiple linear regression or generalized linear mixed models with space as a random effect.
- Finally, the authors could consider using a couple of their lakes as examples to briefly speculate on why differences in catch might be observed in response to the same driver. This might help more strongly incorporate the biology of the systems.

Specific comments:

Line 86: define catchment.

Line 105-106: Here is where a brief plain-language explanation of what a BNM is and why its useful might be extremely helpful.

Line 110: perhaps, here, touch on the important differences between continents that makes this study particularly interesting – to tie in nicely with the later discussion of the apparent benefits of having access to clean water.

Line 114: here the benefit of using diverse lakes is stated, but there is no mention any of the caveats with having such diverse lakes. Because the authors are looking across a global scale, it might be helpful to mention here how this diversity could impact the analyses.

Line 155: not clear how the values for these “adjustments” were chosen

Line 309: “factor can result in an either increase or a decrease...” should be “factor can result in either an increase or a decrease...”

Line 352-354: consider re-phrasing. Confusing presentation of result.

Line 353: spelling: “indicated”, not “indicted”

Reviewer #4 (Remarks to the Author):

The topic of the manuscript is interesting and relevant. The manuscript represents an important application with regards to both environmental aspects and social security and livelihoods that presents interest to the wider community. However, the manuscript needs a lot of work regarding the methodological part. Specifically, the authors need to further clarify and explain the motivation of the Bayesian network use with regression models. The probabilistic formulation of some of the parameters is there and I understand the use of priors with MCMC simulations. However, this does

not mean that a Bayesian network model is applied when the relationships are modelled through regression.

Overall, I find the methods difficult to follow. Specifically, I find confusing the use of Bayesian networks and regression models simultaneously. It is not clear to me the motivation behind both approaches. Could the authors please describe the motivation behind the use of both approaches and specifically, the regression model. I can see better the use of the Bayesian network model with MCMC simulations. There is often jumping between one method to the next in the Methods, so it is hard to follow. Perhaps, the authors could re-structure the Methods in a different way, so that it is easier to follow. Perhaps the sub-sections of the Methods could be re-arranged to align with the use of the different methods (when and how). Maybe, a flowchart of the methodological process would help.

For example, I'm not clear with sentence on lines 442:445, did you just use a Bayesian network to define conceptually the relationships between the different variables? The Bayesian network does not have to be mathematically converted using regression to model the relationships between the different variables. I do not understand what is the motivation to link the variables through regression, if that is the case, where is the Bayesian network application?

Sentence is not clear. Why is the motivation behind converting the BN model to regression models? The use of the word "linking" in "linking BNM variables" is not clear. Similarly, the wording "series of probabilistic models", I find confusing, it is just one Bayesian model? The use of the word model is not correctly used. There is a lot of mixing in the Methods section between regression and probabilistic models. I encourage the author to re-write the section in a clearer way to explain why were the regression models for used and why and similarly for the Bayesian networks?

On line 445: "the structure of our BNM is described here" is not correct when statements such as the sentence on line 443:445 precede this.

Also, can the authors clarify, how can time series data be used with a static Bayesian network formulation?

In the Results section, the authors mention "conceptual model" which implies that the Bayesian network formulation was not done to model the relationships in the study but just to graphically represent the relationships?

The sentence on lines 500:503 is not clear, needs clarification, specifically to the use of the "interdependency" and "were connected by predicted values of lake-environmental variables". I find it very confusing. Similarly, the use of the word probabilistic model is not most correct. Perhaps, you could re-write it to probabilistic formulation or definition, it is only one probabilistic model.

Please, explain in more detail the coefficient estimation on lines: 503:508 and specifically, clarify the coefficient estimation done on a model-by-model basis for the regression part?

Please explain the sentence on lines: 515-518 and specifically, the motivation behind the coefficients for the distribution? The sentence is not very clear at the moment.

Could you please provide an example for the estimation of positive, negative and mixed effects on lines: 532:540? At the moment, the sentence is not clear, and a specific example might help. Also, can you please provide further explanation/clarification why were the effects estimated in this way? Is there a reference you could provide to support this way of estimating the effects?

There is no explanation of the clustering in the Methods section?

Introduction

From line 117:150, the text describing what was done in the methods is not clear. Some of the sentences are very long and difficult to follow. It is hard to get an overall picture of what was done, why and the importance of it. Some of the sentences are too detailed, for example the sentence on line: 141:13. Is this important part, does it relate to the novelty and/or significance of the work? On line 145, the author mentions the use of regression models specifically to “identify socio-economic characteristic...” but from the Methods section, regression models were used throughout and simultaneously with the Bayesian network? It is not clear, again the motivation behind the use of the two approaches, and which methods comes first, where and why?

I find confusing the numbers n Fig.3, are not they results? Is it necessary for these numbers to be shown on top of this graph, are not they a bigger part of the Results?

Can you explain the “stocking” term used for fishing effort?

There is a lot of mixing of terminology throughout the manuscript when it comes to describing the different variables. Maybe a table that shows the different variables and their categories (e.g. climate vs land-use) might help and keeping consistency throughout the text to describe the variables will help to clarify the text.

Overview of the revision

In this document we give our point-by-point responses to all comments from the three reviewers. We used different fonts and colors for reviewers' comments and our responses and noted the line numbers in the revised manuscripts when changes were made. The main changes in the revised manuscript included (1) a major revision and restructuring the Methods section to improve the clarity, (2) adding two tables and revised Fig. 3 to improve the clarity for the Results section, and (3) condensing the Introduction section and adding a few points in the Discussion section as addressed by the reviewers. We believe that our manuscript has improved after these changes in response to the reviewers' comments. We are pleased to submit our revised manuscript.

Response to Reviewer #1s' comments

The authors present a very interesting analysis assessing the interactions between recent climate change, intensification of land use and recorded catch from 31 major freshwater lakes situated across 5 continents. The scope and scale of this study are remarkable and clearly fit within the requirements for this journal. The methods are robust, appropriate for this study and to the best of my knowledge conducted in the correct manor. However, the results, or at least the current presentation of them, are ambiguous and fail to offer a single mechanism describe the environmental drivers and recorded catch. In addition, the paper seems quite long for this journal and I don't feel that 8 large figures are warranted in the main body of the paper.

I outline these thoughts in more detail below. Ultimately, I think that while the authors have a unique dataset with which to examine biotic-abiotic relationship of great global importance.

Bottomline: restructuring of the paper is necessary to allow readers to better interpret their findings.

Bayesian network models are an extremely power tool, well suited to the type of study being performed here. I am not an expert in this method however and my lack in comments regarding the details of the authors methodology should not be interpreted as support for it, I simply do not have the knowledge to adequately critique it and defer to other more experienced reviewers in this regard.

Repeatedly, in the abstract and discussion the authors refer to positive and negative relationships observed between the same factors across the study, but do not provide a clear understanding of why causes this variation in responses. While this may be an accurate reflection of their findings it does not advance the general understanding of these relationships. There must be some factor which accounts for why this relationship is positive or negative.

Given the availability of data regarding these lakes, I would hope that this factor could be identified and included in the models. Conceivably the type of species (pelagic/ benthic; primary consumer/predator; large/small) or the principal methods of fishery in each lake could play a role here. For example, the authors reference several possible factors in the opening paragraphs of the discussion but these are not included in the model. Considering the flexibility of their Bayesian approach this seems like a limitation to the paper.

In any case, I think the authors need to find a way to address this issue for the paper to be suitable for publication in Nature Communications. They have performed a robust analysis of a unique dataset to address a question of ecological, economic importance but the take home message fails to deliver on this opportunity. Should the authors be able to address this, which I think they will, then the paper will be an outstanding contribution to the field and a suitable

publication for this journal.

Response: Reviewer #1's most pointed comment was that we "fail to offer a single mechanism describe the environmental drivers and recorded catch". We empathize with this criticism and obviously have diligently sought the same outcome of identifying a consistent mechanistic explanation for our results. Fig. 8 was our attempt to find some more general pattern, and as Reviewer 3 noted, we believe our finding that catch reductions are less likely to occur in lakes with higher access to clean water is very intriguing and could spur future research to explore possible mechanisms and even encourage developing countries that benefits to clean water protection could have simultaneous benefits to fisheries and food security.

Nonetheless, after reading this review, we further considered how we might restructure our model to evaluate possible hypotheses, including this suggestion to break down the total catch into different guilds or trophic levels in case this would explain why one lake responds in one direction whereas another lake responds in an opposite direction. We face two major obstacles, however. First, we cannot break out fish catch and fishing effort into species or guilds (or trophic level) for all of our lakes, as data are too coarse for at least 18 of the 31 of our study lakes. Second, we considered evaluating these hypotheses for a subset of lakes where the data were available, but then realized that it would require fundamental changes to the Bayesian network model that is currently seeking to predict total catch. For example, a model that is developed to predict catch of planktivorous fishes and catch of piscivorous fishes would require taking trophic interactions into account.

After this discernment, we realized that this suggested approach was beyond the scope of our data, as well as the goal of our current analyses which we stated was to determine

whether “changes in climate and land use affected lake fish catches across the globe”. We feel there is high value in evaluating and reporting whether such general patterns exist across lakes and argue that evaluating more specific patterns (e.g., why climate and land-use changes increase the proportion of planktivorous fishes in a fishery across global lakes) might be a next research step. Nonetheless, to partially accommodate this specific reviewer concern, we added text in the Discussion section (in the paragraph starting from line 248) that highlights the need for future research to understand why the relationship between a climate or land-use driver and fish catch was either positive or negative and suggest that exploring differential composition of the catch for lakes where the data exist could be a logical starting point.

Regarding the reviewer’s comments about the length of this manuscript, we believe that this manuscript is not too long based on author guideline of *Nature Communications*.

Response to Reviewer #3s' comments:

Summary

The authors report a global analysis of the effects of climate and land use changes on fisheries catches via their effects on lake environments. Their key findings suggest that climate and land use have negative impacts on catch (i.e. decrease in catch), but as a result of changes to the lake environment that occur in either direction (e.g. warmer or colder air temperatures, more or less precipitation, higher or lower chlorophyll-a). Further, the authors find that a reduction in catch is less likely to occur in cleaner lakes, thus highlighting the need to increase water quality in developing countries to simultaneously benefit drinking water and fisheries health under future change.

Overall, the study was interesting and analyzes an impressive data set. The findings seem to be novel and should be of wide interest. The authors use their analysis to draw very general, but important conclusions that link the effects of climate change on lake ecosystems to socio-economic issues. I also appreciated the transparency of the study, as the authors provided all data and code used for the analyses. Below are some comments that hopefully the authors find helpful and could help strengthen the manuscript.

General comments

- From reading the abstract, I found that one of the most powerful claims the authors make is how water cleanliness can affect how vulnerable a fishery is to changes to climate/land-use. However, it appears that clean lakes are more common in developed countries which would also have more policies to prevent over-fishing. How does the analysis decouple higher drinking water quality from higher fishing regulations in more developed countries, which could also buffer these systems from significant decreases in catch?

Response: Excellent point. We now address this point in discussion (line 333) by making it clear that lakes in areas with more access to clean water usually also have stronger fishing regulations, which could also make fish catches less vulnerable to effects of climate and land use changes by maintaining healthy fish stocks. To our knowledge, there are some types of well-enforced fishing regulations across the 14 study lakes with >85% of catchment population having access to clean water, which suggests that the strength of fishing regulations could be positively correlated with the access to clean water across lakes. However, we could not quantitatively decouple a possible covariation between access to clean water and fishing

regulations because we did not have lakes with high access to clean water and minimal or no fishing regulations.

- Is catch calculated as a community estimate? If so, is there a size limit for each fishery? Since more developed countries tend to be more discriminate in their fishing, the authors could be more specific on the size ranges that were used to determine catchability coefficients, and thus to estimate total catch for each site. How, for example, did the authors account for lakes in North America and Europe potentially having multiple, yet targeted, fisheries in each lake versus non-targeted or lack of length regulations in African or Asian fisheries?

Response: Fish catch in this study is defined as the total of all fish species caught on an annual basis. The data for fish catches and fishing effort used in this study were statistics reported by research or management agencies (Supplementary Table S3). For some fishes in some of the 31 study lakes, there were undoubtedly size limits for some specific fisheries and some multi-species fisheries, but we could not easily account for this in our model.

We argue that accounting for an index of fishing effort, alone, was not only critical to our model but also a relatively complicated process given the diversity of fishery types and effort that has been executed on these lakes. As we described in the Methods section (in the paragraph starting from line 442), the assumption underlying the conditional probability distribution for fish catch in our BNM was that there is a lake-specific, empirical relationship between fishing catchability and fishing effort and fish biomass. Standardizing effort across these lakes and fishery types was a difficult task, and we would be concerned about over-parameterization of the model had we attempted to model the size distributions of populations and then estimated catchability for each fishery type on each population as suggested.

- Because the sites are located across continents and some tend to be more clustered than others, I wonder if the analysis would benefit from testing the effects of spatial autocorrelation. I'd imagine that over time, sites within close geographical range would have similar climates and land use, so I wonder how reasonable it is to assume that they are independent when trying to unpack the climate versus land use versus socio-economic impacts on fisheries catch. For example, each region might have very different end-points of in terms of current or future changes to each class of variables tested (e.g. the least clean lakes in developed countries might be cleaner than the cleanest of those in developing countries, or differences in seasonality, photoperiod, etc.).

Response: Yes, we agree that clustered lakes might have had similar climate forcing (or even land use) factors. When we look at our results, however, we see specific examples of clustered lakes did not necessarily respond in a similar direction to climate and land use drivers. For example, lakes Erie, Michigan, Oneida, and Simcoe can be considered within the same region of North America (Fig. 1) but our results showed that fish catches in these four lakes responded to climate and land-use changes very differently (Figs. 4–6). The colors of the bars on these figures also demonstrate the lack of consistent patterns within a continent.

- Not being an expert with the BNM approach used, I think the manuscript would benefit from a brief mention of what the BNM is, what it does, and why it was chosen above a more simple method of performing a multiple linear regression or generalized linear mixed models with space as a random effect.

Response: We added an opening sub-section in the Methods section to explain why we chose the Bayesian networks modeling approach (lines 377–395) and added a plain-language explanation in the Introduction section as the Reviewer suggested in Specific comments (lines 112–120).

- Finally, the authors could consider using a couple of their lakes as examples to briefly speculate on why differences in catch might be observed in response to the same driver. This might help more strongly incorporate the biology of the systems.

Response: We appreciate this author suggestion, and went back and forth (even before our first submission) on whether to delve into explaining lake-specific patterns. In the end, we decided against it because the goal of the paper was to evaluate whether general trends can be detected across global lakes, and we hesitated to cherry-pick a few lakes to devote more in-depth discussion or explanation. Rather, we offered speculations for why fish catches responded differently to the same driver across lakes (in the 2 paragraphs starting from line 248 and line 346).

Specific comments

Line 86: define catchment.

Response: We made it clear catchment here means hydrological catchment or drainage basin (line 101).

Line 105-106: Here is where a brief plain-language explanation of what a BNM is and why its useful might be extremely helpful.

Response: We added text explaining why the modeling approach was chosen and what a BNM is (lines 112–120).

Line 110: perhaps, here, touch on the important differences between continents that makes this study particularly interesting – to tie in nicely with the later discussion of the apparent benefits of having access to clean water.

Response: We added a sentence to point out that lakes in Europe and Americas generally have much better socio-economic environment than lakes in Asia and Africa (line 124).

Line 114: here the benefit of using diverse lakes is stated, but there is no mention any of the caveats with having such diverse lakes. Because the authors are looking across a global scale, it might be helpful to mention here how this diversity could impact the analyses.

Response: The caveats are associated with data limitations and lake-by-lake differences. We have used a whole paragraph to address these issues in the Discussion section (starting from line 346). Perhaps, there is no need to address these issues again in the Introduction section.

Line 155: not clear how the values for these “adjustments” were chosen

Response: We added a sentence to address that our results have accounted for fish stocking and fishing efforts, which both could have strong effects on fish catches (line 150).

Line 309: “factor can result in an either increase or a decrease...” should be “factor can result in either an increase or a decrease...”

Response: This sentence has been revised as address Reviewer #1’s comments.

Line 352-354: consider re-phrasing. Confusing presentation of result.

Response: We revised the sentence (lines 294–297)

Line 353: spelling: “indicated”, not “indicted”

Response: We corrected the mistake and read through the manuscript a few times to ensure that there is no spelling mistake.

Response to Reviewer #4s' comments:

The topic of the manuscript is interesting and relevant. The manuscript represents an important application with regards to both environmental aspects and social security and livelihoods that presents interest to the wider community.

However, the manuscript needs a lot of work regarding the methodological part. Specifically, the authors need to further clarify and explain the motivation of the Bayesian network use with regression models. The probabilistic formulation of some of the parameters is there and I understand the use of priors with MCMC simulations. However, this does not mean that a Bayesian network model is applied when the relationships are modelled through regression.

Overall, I find the methods difficult to follow. Specifically, I find confusing the use of Bayesian networks and regression models simultaneously. It is not clear to me the motivation behind both approaches. Could the authors please describe the motivation behind the use of both approaches and specifically, the regression model. I can see better the use of the Bayesian network model with MCMC simulations. There is often jumping between one method to the next in the Methods, so it is hard to follow. Perhaps, the authors could re-structure the Methods in a different way, so that it is easier to follow. Perhaps the sub-sections of the Methods could be re-arranged to align with the use of the different methods (when and how). Maybe, a flowchart of the methodological process would help.

For example, I'm not clear with sentence on lines 442:445, did you just use a Bayesian network to define conceptually the relationships between the different variables? The Bayesian network does not have to be mathematically converted using regression to model the

relationships between the different variables. I do not understand what is the motivation to link the variables through regression, if that is the case, where is the Bayesian network application?

Sentence is not clear. Why is the motivation behind converting the BN model to regression models? The use of the word “linking” in “linking BNM variables” is not clear. Similarly, the wording “series of probabilistic models”, I find confusing, it is just one Bayesian model? The use of the word model is not correctly used. There is a lot of mixing in the Methods section between regression and probabilistic models. I encourage the author to re-write the section in a clearer way to explain why were the regression models for used and why and similarly for the Bayesian networks?

Response: Thank you for these suggestions. We revised our Methods section to improve the clarity. Specifically, we added an opening sub-section to explain why we chose the Bayesian networks modeling approach (lines 377–395), rewrote the sub-section for model development (lines 396–457), and separated the derivation of prior distributions part as an independent sub-section from the coefficient estimation sub-section and gave more details to address the Reviewer’s comments there (lines 458–475).

As the Reviewer pointed out, our uses of the terms “regression model(s)” and “probabilistic model’s” are both confusing throughout the manuscript so we replaced all of them with more precise descriptions of what we used or did in the revised manuscript. For example, we replaced “estimated by regression models” with “least-squares estimates” in the new subsection for prior distributions (lines 458–475) and replaced “regression lines” with “least-squares lines” in Fig. 7.

On line 445: “the structure of our BNM is described here” is not correct when statements such as the sentence on line 443:445 precede this.

Response: This sentence was deleted in the revision.

Also, can the authors clarify, how can time series data be used with a static Bayesian network formulation?

Response: Generally, it is because we assumed that annual time-series data are independent observations, such that the value of each response variable in one year is dependent upon values of predictor variables of the model in the same year but independent from all of its values in the other years. We gave a detail explanation in the revised subsection for model development (lines 396–410).

In the Results section, the authors mention “conceptual model” which implies that the Bayesian network formulation was not done to model the relationships in the study but just to graphically represent the relationships?

Response: We did formulate and use a Bayesian Networks Model. The term “conceptual model” was used in the caption of Fig. 3 and the panel h. of Figs. 4–6, where we qualitatively summarized our simulation results across the 31 study lakes. As we revised the Fig. 3 following another suggestion by the Reviewer, the term was no longer used in the caption. In Figs. 4–6, we replace the term with “summary”, which better describes what the panel h. in of Figs. 4–6 is for.

The sentence on lines 500:503 is not clear, needs clarification, specifically to the use of the

“interdependency” and “were connected by predicted values of lake-environmental variables”. I find it very confusing. Similarly, the use of the word probabilistic model is not most correct. Perhaps, you could re-write it to probabilistic formulation or definition, it is only one probabilistic model.

Response: This sentence has been deleted in the revised manuscript as we re-wrote the subsection for model development.

Please, explain in more detail the coefficient estimation on lines: 503:508 and specifically, clarify the coefficient estimation done on a model-by-model basis for the regression part? Please explain the sentence on lines: 515-518 and specifically, the motivation behind the coefficients for the distribution? The sentence is not very clear at the moment.

Response: We separated the derivation of prior distributions part as an independent subsection from the coefficient estimation sub-section and explained why we used prior multivariate normal distributions for Bayesian networks model coefficients (lines 458–475).

Could you please provide an example for the estimation of positive, negative and mixed effects on lines: 532:540? At the moment, the sentence is not clear, and a specific example might help. Also, can you please provide further explanation/clarification why were the effects estimated in this way? Is there a reference you could provide to support this way of estimating the effects?

Response: We changed the subheading from “Model interpretation” to “Effects between variable pairs” and revise the subsection accordingly to improve the clarity and provide explanation why the effects were estimated in this way (lines 487–494).

There is no explanation of the clustering in the Methods section?

Response: We used the term once in the caption of Fig. 8. Because it was a descriptive, rather than statistical term, in our use, we replaced it with “distribution” to reduce confusion.

Introduction

From line 117:150, the text describing what was done in the methods is not clear. Some of the sentences are very long and difficult to follow. It is hard to get an overall picture of what was done, why and the importance of it. Some of the sentences are too detailed, for example the sentence on line: 141:13. Is this important part, does it relate to the novelty and/or significance of the work?

Response: We condensed the three paragraphs to better present the big picture of our analysis (lines 129–138).

On line 145, the author mentions the use of regression models specifically to “identify socio-economic characteristic...” but from the Methods section, regression models were used throughout and simultaneously with the Bayesian network? It is not clear, again the motivation behind the use of the two approaches, and which methods comes first, where and why?

Response: This should be better stated as a “correlation analysis” following our BNM simulations. We revised the language accordingly throughout the revised manuscript.

I find confusing the numbers in Fig.3, are not they results? Is it necessary for these numbers to be shown on top of this graph, are not they a bigger part of the Results?

Response: We separated results part from Fig. 3 so that the revised Fig. 3 has a focus of presenting the hypothesized causal processes and added Table 2 to report these results.

Can you explain the “stocking” term used for fishing effort?

Response: Stocking is a practice to enhance fish biomass by releasing hatchery-raised fish into lakes. It is independent from fishing effort in the model. We changed the term “stocking” to “fish stocking” when appropriate in the text and in Fig. 3 to avoid confusion.

There is a lot of mixing of terminology throughout the manuscript when it comes to describing the different variables. Maybe a table that shows the different variables and their categories (e.g. climate vs land-use) might help and keeping consistency throughout the text to describe the variables will help to clarify the text.

Response: We added Table 1 to fit this need.

Reviewers' comments:

Reviewer #1 (Remarks to the Author):

The authors revisions have adequately addressed my concerns. It is unfortunate that the data cannot not provide a clearer, or at least simpler, single explanation for the trends observed, but ecology is rarely that simple to explain. The manuscript is now suitable for publication in my view.

Reviewer #5 (Remarks to the Author):

I was asked to review the methods of the paper in this revised paper, so that is where I'm focusing.

The methods are rather difficult to follow - a flow chart or similar would be very useful as proposed by one of the other reviewers.

I gather you made separate models (or a model structure that does effectively the same) for each of the 31 lakes, to be able to produce lake-specific results? I may have misunderstood the methods section, but I take it that you modelled the beta parameters for all lakes separately, where data allowed, then combined these into a MVN distribution and used it as a prior for every individual lake model. Is that correct? In that case, when a lake is missing data from a parameter, the prior would not be updated, and in effect, the parameters for that model would just be the prior, i.e. the mean of the lakes where the data does exist. Since these lakes are very different from each other, I wonder how useful this approach is - there is no discussion on this save for a brief mention of uncertainties related to individual lakes. I don't believe parameters such as how CHL depends on PRE and LUag can be transferred from one lake to another.

I am worried about a few other things in the data use as well. The relationships between the variables are modelled as linear or log-log-linear, although some of them may, in reality, have nonlinearities such as step functions, regime shifts etc. The model will behave according to these linearity assumptions, but the results may not be realistic at all. At least, these linearities should be justified, perhaps changed.

I am also wondering about the catchability. It is well-known that it varies drastically between species, but this model seems to model one catchability for all species per lake, at least, and even learn (via the prior) across lakes.

Why did you choose to do the assessment "bottom-up", i.e. simulating the low catches and seeing how the parents change, instead of doing it in the causal direction? I could see some reasons, but I'd like to see this approach motivated.

Towards the end of the paper, there is some discussion of the consequences of the predicted climate change. I do not believe this model should be used to extrapolate outside of the range of climate parameters that have been used when learning the model. Again, if the world were linear, it could, but there may be multiple processes that are nonlinear and therefore the extrapolation is not warranted. For example, a fish species or community may have been able to adapt to the current temperature increase so far, but may be close to its tolerance limits and may collapse once the temperature rises a little more.

Overview of the revision

In this document, we give our point-by-point responses to the comments on the methods used in this study from Reviewer #5. Once again, we used different fonts and colors for the reviewer's comments and our responses and highlighted changes made in this revision (in red) throughout the main manuscript. The main change in the revised manuscript was adding a flow chart (Fig. 3) that summarizes the procedures of three analyses conducted in this study. Due to the limited number (10) of display items (figures and tables), we moved Fig. 2 "Fish catches across the 31 study lakes in the period 1970–2014" from the main text to Supplementary Information (SI). We believe that Fig. 2 better fits SI because it displays data reported in SI and there was no discussion of Fig. 2 in the main text. Additionally, we also made a few minor edits, which were all highlighted, to improve the clarity. We are pleased to submit our revised manuscript.

Response to Reviewer #5s' comments

I was asked to review the methods of the paper in this revised paper, so that is where I'm focusing.

The methods are rather difficult to follow - a flow chart or similar would be very useful as proposed by one of the other reviewers.

Response: We added a flow chart (Fig. 3) that summarizes the procedures of three analyses conducted in this study.

I gather you made separate models (or a model structure that does effectively the same) for each of the 31 lakes, to be able to produce lake-specific results? I may have misunderstood

the methods section, but I take it that you modelled the beta parameters for all lakes separately, where data allowed, then combined these into a MVN distribution and used it as a prior for every individual lake model. Is that correct? In that case, when a lake is missing data from a parameter, the prior would not be updated, and in effect, the parameters for that model would just be the prior, i.e. the mean of the lakes where the data does exist.

Response: The understanding for the derivation of our prior distributions is correct. However, the scenario “when a lake is missing data from a parameter, the prior would not be updated, and in effect, the parameters for that model would just be the prior, i.e. the mean of the lakes where the data does exist” did not exist in this study. For all variables in our Bayesian networks model (BNM), we were able to obtain data to derive model inputs for at least nine years for each of the 31 study lakes (refer to Supplementary Table S3 for data availability). Therefore, the prior distributions of the BNM coefficients were always lake-specific and the posterior distributions of the BNM coefficients were always developed based on pooled information from the lake and the other 30 lakes. We made it clear that prior distributions of the BNM coefficients were always lake-specific in Methods (lines 487–490).

Since these lakes are very different from each other, I wonder how useful this approach is - there is no discussion on this save for a brief mention of uncertainties related to individual lakes. I don't believe parameters such as how CHL depends on PRE and LUag can be transferred from one lake to another.

Response: We understand the Reviewer's concerns about the “transferability” of information across lakes and the fact that the estimation of our lake-specific BNM coefficients were influenced by information across all of the 31 study lakes. While it would be ideal if the

information across the lakes was transferrable (which would lower the required lake-specific sample size to reduce the estimation uncertainty), we had no prior assumption for information transferability across the lakes. Rather, our BNM was structured in a way to have the transferability of information across the lakes informed by data. In the derivation of posterior distributions of our BNM coefficients, the relative importance of lake-specific information increases with lake-specific sample size and across-lake difference. For example, if a lake was very different from all the other lakes, the relative importance of lake-specific information would be much higher than across-lake information in the derivation of posterior distributions of lake-specific BNM coefficients. In contrast, if all lakes were similar in some aspect, the across-lake information would reduce the estimation uncertainty in the estimation of lake-specific BNM coefficients. We added a paragraph to clarify how lake-specific and across-lake information was used in the derivation of posterior distributions of our BNM coefficients in Methods (lines 493–501).

I am worried about a few other things in the data use as well. The relationships between the variables are modelled as linear or log-log-linear, although some of them may, in reality, have nonlinearities such as step functions, regime shifts etc. The model will behave according to these linearity assumptions, but the results may not be realistic at all. At least, these linearities should be justified, perhaps changed.

Response: We believe that it is justifiable to use the linear or log-log linear functions in our BNM, as they have been well studied and used frequently in previous studies. The primary

literature that justified the use of each function has been given in *Model development in Methods*.

I am also wondering about the catchability. It is well-known that it varies drastically between species, but this model seems to model one catchability for all species per lake, at least, and even learn (via the prior) across lakes.

Response: The Reviewer is right that catchability could vary between species. Nonetheless, a model that estimates catchability for each species in each lake was beyond the capacity of our data, which are too coarse for us to break out fish catch and fishing effort into species level for all of our 31 study lakes. To account for effects of fishing effort on total fish catch in our BNM, our approach was to model the (weighted-)mean catchability across all species in each lake. As shown in equation (8), we modeled the mean catchability across all species as a function of total fish biomass and total fishing effort.

Why did you choose to do the assessment "bottom-up", i.e. simulating the low catches and seeing how the parents change, instead of doing it in the causal direction? I could see some reasons, but I'd like to see this approach motivated.

Response: Good question. As stated in the Methods and more clearly illustrated in the new Fig. 3, we simulated how catches would respond when randomly sampling climate and land use drivers (to avoid extrapolation) and the BNM coefficients. In this approach (which we assume you mean is "bottom-up"), our only assumption was what level of % catch decrease we would report, and we justified this in lines 526–529: "We used a 25% catch decrease as our simulation target because it is close to the maximum value of standard errors for the ratios of fish catch to

median catch, which ranged from 2.1% to 25.7% across the study lakes in the study period.” An alternative approach would have been to systematically vary the drivers in some alternative way (rather than the random sampling) and report how the specific change (e.g., 15% increase) in catch (we assume this is what you meant by “top-down”). We ultimately decided against this approach because it required us to justify how we varied each of the three drivers (i.e., air temperature, precipitation, agricultural land use). We believe that making fewer assumptions could avoid further complicating the already complicated methods of this study.

Towards the end of the paper, there is some discussion of the consequences of the predicted climate change. I do not believe this model should be used to extrapolate outside of the range of climate parameters that have been used when learning the model. Again, if the world were linear, it could, but there may be multiple processes that are nonlinear and therefore the extrapolation is not warranted. For example, a fish species or community may have been able to adapt to the current temperature increase so far, but may be close to its tolerance limits and may collapse once the temperature rises a little more.

Response: Excellent point. We have now added text to caution against extrapolation outside the range of climate or land use inputs that were used in our model (lines 276–279, 303–306, and 330–333).

REVIEWERS' COMMENTS:

Reviewer #5 (Remarks to the Author):

Thank you to the authors for their constructive approach to the comments, and clarifications in the manuscript. I believe the paper can be now published.

Just one thing: on rows 493-495: "While the prior distributions of our BNM coefficients were 494 developed on a lake-by-lake basis, the posterior distributions of our BNM coefficients were 495 developed based on pooled information across the 31 study lakes." Prior and posterior should be the other way around in the first sentence, right?

Yours,

Laura Uusitalo, PhD

Finnish Environment Institute

Response to Reviewer #5s' comments

Thank you to the authors for their constructive approach to the comments, and clarifications in the manuscript. I believe the paper can be now published.

Just one thing: on rows 493-495: "While the prior distributions of our BNM coefficients were developed on a lake-by-lake basis, the posterior distributions of our BNM coefficients were developed based on pooled information across the 31 study lakes." Prior and posterior should be the other way around in the first sentence, right?

Yours,

Laura Uusitalo, PhD

Finnish Environment Institute

Response: We are thankful to Dr. Laura Uusitalo's effort and comments. About the final comment, we also found the first part of this sentence confusing now so we deleted it and revised the opening of this section (line 496) as

"Coefficient estimation. The posterior distributions of our BNM coefficients were developed based on pooled information across the 31 study lakes. Our BNM was..."